# Genomic language model predicts protein co-regulation and function

Yunha Hwang ®[1] ✉, Andre L. Cornman[2], Elizabeth H. Kellogg[3,5], Sergey Ovchinnikov ®[4,6] ✉ & Peter R. Girguis ®[1] ✉

Deciphering the relationship between a gene and its genomic context is fundamental to understanding and engineering biological systems. Machine learning has shown promise in learning latent relationships underlying the sequence-structure-function paradigm from massive protein sequence datasets. However, to date, limited attempts have been made in extending this continuum to include higher order genomic context information. Evolutionary processes dictate the specificity of genomic contexts in which a gene is found across phylogenetic distances, and these emergent genomic patterns can be leveraged to uncover functional relationships between gene products. Here, we train a genomic language model (gLM) on millions of metagenomic scaffolds to learn the latent functional and regulatory relationships between genes. gLM learns contextualized protein embeddings that capture the genomic context as well as the protein sequence itself, and encode biologically meaningful and functionally relevant information (e.g. enzymatic function, taxonomy). Our analysis of the attention patterns demonstrates that gLM is learning co-regulated functional modules (i.e. operons). Our findings illustrate that gLM's unsupervised deep learning of the metagenomic corpus is an effective and promising approach to encode functional semantics and regulatory syntax of genes in their genomic contexts and uncover complex relationships between genes in a genomic region.

Evolutionary processes result in the linkage between protein sequences, structure and function. The resulting sequence-structure-function paradigm[1] has long provided the basis for interpreting vast amounts of genomic data. Recent advances in neural network (NN)-based protein structure prediction methods[2,3], and more recently protein language models (pLMs)[4–7] suggest that data-centric approaches in unsupervised learning can represent these complex relationships shaped by evolution. To date, these models largely consider each protein as an independent and standalone entity. However, proteins are encoded in genomes alongside other proteins, and the specific genomic context that a protein occurs in is determined by evolutionary processes where

each gene gain, loss, duplication and transposition event is subject to selection and drift[8–10]. These processes are particularly pronounced in bacterial and archaeal genomes where frequent horizontal gene transfers (HGT) shape genomic organization and diversity[11,12]. Thus, there exists an inherent evolutionary linkage between genes, their genomic context, and gene function[13–15], which can be explored by characterizing patterns that emerge from large metagenomic datasets.

Recent efforts to model genomic information have shown predictive power of genomic context in gene function[16] and metabolic trait evolution[17] in bacterial and archaeal genomes. However, these methods represent genes as categorical entities, despite these genes

[1]Department of Organismic and Evolutionary Biology, Harvard University, Cambridge, MA, USA. [2]Tatta Bio, Baltimore, MD, USA. [3]Department of Molecular Biology and Genetics, Cornell University, Ithaca, NY, USA. [4]John Harvard Distinguished Science Fellowship Program, Harvard University, Cambridge, MA, USA. [5]Present address: Department of Structural Biology, St. Jude Children's Research Hospital, Memphis, TN, USA. [6]Present address: Department of Biology, Massachusetts Institute of Technology, Cambridge, MA, USA. ✉e-mail: yhwang@g.harvard.edu; so3@mit.edu; pgirguis@oeb.harvard.edu

existing in continuous space where multidimensional properties such as phylogeny, structure, and function are abstracted in their sequences. On the other end of the spectrum of representations, there have been efforts to use unsupervised learning on nucleotide sequences to predict gene expression level[18] and detect regulatory motifs[19–21]. These models are largely trained and benchmarked on the human genome and focus on predicting gene regulation rather than function. Most recent efforts to leverage diverse microbial sequences to model genome-scale information include GenSLMs[22], which is pretrained on codon-level representations of diverse bacterial and viral gene sequences and later fine-tuned on SARS-CoV-2 genomes. In order to learn generalizable gene-to-gene-context interactions across biology, a model needs to be pretrained on 1) diverse lineages of organisms, 2) rich and continuous representation of genes and 3) longer segments of genomes with multiple genes. To our knowledge, there has been no method that combines all three aspects of pretraining to learn genomic information across diverse lineages of biology (see summary of previous studies in Supplementary Table 1).

In order to close the gap between genomic-context and gene sequence-structure-function, we develop a genomic language model (gLM) that learns contextual representations of genes. gLM leverages pLM embeddings as input, which encode relational properties[4] and structure information[23] of the gene products. Our model is based on the transformer[24] architecture and is trained using millions of unlabelled metagenomic sequences via the masked language modeling[25] objective, with the hypothesis that its ability to attend to different parts of a multi-gene sequence will result in the learning of gene functional semantics and regulatory syntax (e.g. operons). Here, we report evidence of the learned contextualized protein embeddings and attention patterns capturing biologically relevant information. We demonstrate gLM's potential for predicting gene function and co-regulation, and propose future applications and research directions, including transfer learning capabilities of gLM.

## Results

### Masked language modeling of genomic sequences

Language models, such as Bidirectional Encoder Representations from Transformers (BERT[25]), learn the semantics and syntax of natural languages using unsupervised training of a large corpus. In masked language modeling, the model is tasked with reconstructing corrupted input text[25], where some fraction of the words are masked. Significant advances in language modeling performance was achieved by adopting the transformer[24] neural network architecture, where each token (i.e. word) is able to attend to other tokens. This is in contrast to Long-Short-Term-Memory networks (LSTMs)[26] that sequentially processes tokens. To model genomic sequences, we trained a 19-layer transformer model (Fig. 1A; for a detailed figure see Supplementary Fig. 1) on seven million metagenomic contig fragments consisting of 15 to 30 genes from the MGnify[27] database. Each gene in a genomic sequence is represented by a 1280 feature vector (context-free protein embeddings) generated by using ESM2 pLM[23], concatenated with an orientation feature (forward or backward). For each sequence, 15% of genes are randomly masked, and the model learns to predict the masked label using the genomic context. Based on the insight that more than

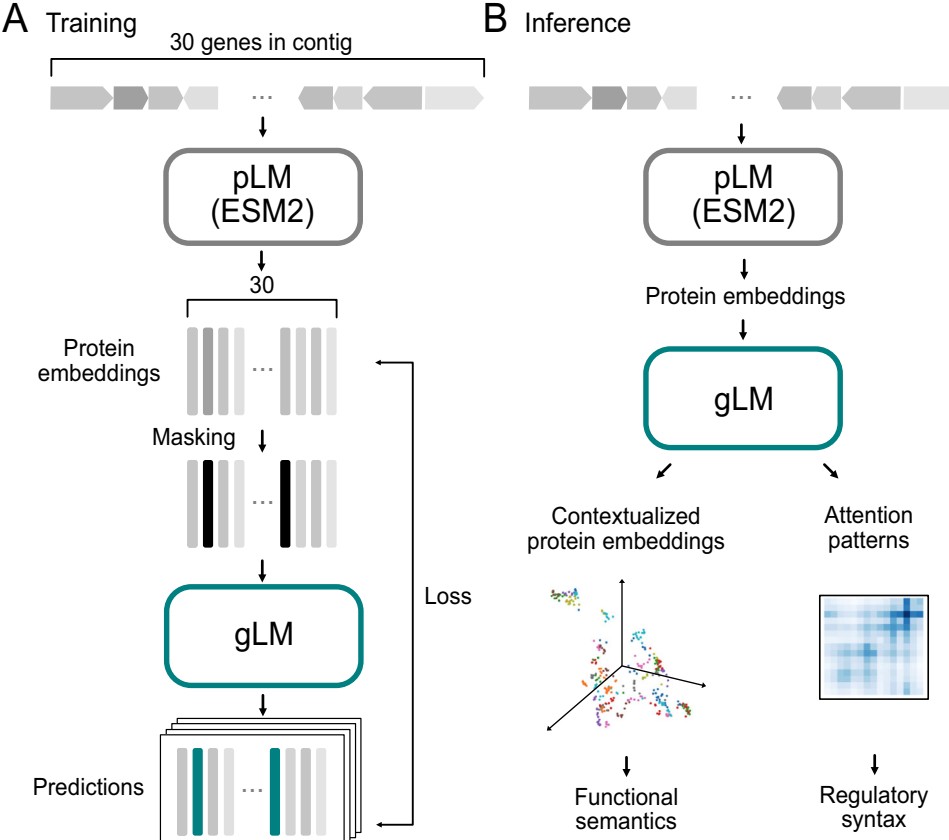

**Fig. 1 | gLM training and inference schematics. A** For training, contigs (contiguous genomic sequences) containing up to 30 genes are first translated into proteins, which are subsequently embedded using a protein language model (pLM) encoder (ESM2). Masked inputs are generated by random masking at 15% probability and genomic language model (gLM; a transformer encoder) is trained to make four predictions for each masked protein, with associated likelihoods.

Training loss is calculated on both the prediction and likelihoods. **B** At inference time, inputs are generated from a contig using ESM2 output. Contextualized protein embeddings (hidden layers of gLM) and attention patterns are used for various downstream tasks. See Supplementary Fig. 1 for detailed schematics. Source data are provided as a Source Data file.

one gene can legitimately be found in a particular genomic context, we allow the model to make four different predictions and also predict their associated probabilities (Supplementary Fig. 1). Thus, instead of predicting their mean value, the model can approximate the underlying distribution of multiple genes that can occupy a genomic niche. We assess the model's performance using a pseudo-accuracy metric, where a prediction is considered correct if it is closest to the masked protein in euclidean distance compared to the other proteins encoded in the sequence (see Methods). We validate our model's performance on the *Escherichia coli* K-12 genome[28] by excluding from training 5.1% of MGnify subcontigs in which more than half of the proteins are similar (>70% sequence identity) to *E. coli* K-12 proteins. It is important to note that our goal was not to remove all *E. coli* K-12 homologs from the training, which would have removed a vast majority of training data as many essential genes are shared across organisms. Instead, our goal was to remove as many *E.coli* K-12-like genomic contexts (subcontigs) from training, which is more appropriate for the training objective. gLM achieves 71.9% in validation pseudo-accuracy and 59.2% in validation absolute accuracy (Supplementary Fig. 2). Notably, 53.0% of the predictions made during validation are with high confidence (with prediction likelihood > 0.75), and 75.8% of the high confidence predictions are correct, indicating gLM's ability to learn a confidence metric that corresponds to increased accuracy. We baseline our performance with a bidirectional LSTM model trained using the same language modeling task on the same training dataset, where validation performance plateaus at 28% pseudo-accuracy and 15% absolute accuracy (Supplementary Fig. 2 and Supplementary Table 2, note that biLSTM is smaller because it failed to converge when scaling the number of layers). We ablate the use of pLM representations as input to gLM by replacing them with one-hot amino acid representations (Supplementary Table 3) and report performance equivalent to random predictions (3% pseudo-accuracy and 0.02% absolute accuracy).

## Contextualized gene embeddings capture gene semantics

The mapping from gene to gene-function in organisms is not one-to-one. Similar to words in natural language, a gene can confer different functions[29] depending on its context[30], and many genes confer similar functions (i.e. convergent evolution[31], remote homology[32]). We used gLM to generate 1280-feature contextualized protein embeddings at inference time (Fig. 1B), and we examined the "semantic" information captured in these embeddings. Analogous to how words are likely to have different meanings depending on the type of text in which they are found (Fig. 2A), we find that contextualized protein embeddings of genes that appear across multiple environments (biomes) tend to cluster based on biome types. We identified 31 proteins in our training database (MGYPs) that occurred more than 100 times and distributed with at least 20 occurrences in each "Host-associated", "Environmental", and "Engineered" biomes according to MGnify's designation. We find that gLM's contextualized protein embeddings capture biome information for the majority ($n = 21$) of these multi-biome MGYPs. For instance, a gene encoding a protein annotated "translation initiation factor IF-1" occurs multiple times across biomes. While the input to gLM (context-free protein embedding; ESM2 representation) is identical across all occurrences, gLM's output (contextualized protein embeddings) cluster with biome types (Fig. 2B; silhouette score = 0.17, see the other 30 multi-biome MGYP visualizations in Supplementary Fig. 3). This suggests that the diverse genomic contexts that a gene occupies are specific for different biomes, implying biome-specific gene semantics.

We explored an ecologically important example of genomic "polysemy" (multiple meanings conferred by the same word; Fig. 2C) of methyl-coenzyme M reductase (MCR) complex. The MCR complex is able to carry out a reversible reaction (Reaction 1 in Fig. 2D), whereby the forward reaction results in the production of methane (methanogenesis) while the reverse results in methane oxidation

(methanotrophy). We first examine the McrA (methyl-coenzyme M reductase subunit alpha) protein in diverse lineages of ANME (ANaerobic MEthane oxidizing) and methanogenic archaeal genomes. These archaea are polyphyletic and occupy specific ecological niches. Notably, similar to how a semantic meaning of a word exists on a spectrum and a word can have multiple semantically appropriate meanings in a context (Fig. 2C), the MCR complex can confer different functions depending on the context. Previous reports demonstrate the capacities of ANME (ANME-2 in particular) carrying out methanogenesis[33] and methanogens conducting methane oxidation in specific growth conditions[34]. The context-free ESM2 embedding of these proteins (Fig. 2E) shows little organization, with little separation between ANME-1 and ANME-2 McrA proteins. However, contextualized gLM embeddings (Fig. 2F) of the McrA proteins show distinct organization where ANME-1 McrA proteins form a tight cluster, while ANME-2 McrA proteins form a cluster with methanogens (silhouette score after contextualization: 0.24; before contextualization: 0.027). This organization reflects the phylogenetic relationships between the organisms that McrAs are found in, as well as the distinct operonic and structural divergence of MCR complexes in ANME-1 compared to those found in ANME-2 and methanogens[35]. As proposed by Shao et al.[35]., the preferred directionality in Reaction 1 (Fig. 2D) in ANME-2 and some methanogens may be more dependent on thermodynamics.

We also demonstrate that contextualized gLM embeddings are more suitable for determining the functional relationship between gene classes. Analogous to how the words "dog" and "cat" are closer in meaning relative to "dog" and "train" (Fig. 2G), we see a pattern where Cas1- and Cas2-encoding genes appear diffuse in multiple subclusters in context-free protein embedding space (Fig. 2H) cluster in contextualized embedding space (Fig. 2I). This reflects their similarity in function (e.g. phage defense). This is also demonstrated in biosynthetic genes, where genes encoding lipopolysaccharide synthase (LPS) and polyketide synthase (PKS) cluster closer together in contextualized embedding space distinct from the Cas proteins (Fig. 2I). We quantitate this pattern with a higher silhouette score measuring phage defense and biosynthetic gene separation (gLM representation: $0.123 \pm 0.021$, pLM representation: $0.085 \pm 0.007$; paired t-test, t-statistic: 5.30, two-sided, $p$ value = 0.0005, $n = 10$). Contextualized protein embeddings are therefore able to capture relational properties akin to semantic information[36], where genes encoding proteins that are more similar in their function are found in similar genomic contexts.

In order to quantify the information gain as a result of training a transformer on genomic contexts, we compare clustering results in 2B, F, and I with clustering conducted on (sub)contig-averaged pLM embeddings (Supplementary Fig. 4). By mean-pooling pLM embeddings across a given subcontig, we can summarize the context information as a naive baseline. We report a most consistent clustering (higher silhouette scores) of gLM embeddings compared to contig-averaged pLM for all three analyses (see Supplementary Fig. 4 figure captions for values). We demonstrate that the gLM transformer model learns representations that correlate with biological function, which are not captured by the naive baseline.

## Characterizing the unknown

Metagenomic sequences feature many genes with unknown or generic functions, and some are so divergent that they do not contain sufficient sequence similarity to the annotated fraction of the database[37]. In our dataset, of the 30.8 M protein sequences, 19.8% could not be associated with any known annotation (see Methods), and 27.5% could not be associated with any known Pfam domains using a recent deep learning approach (ProtENN)[38]. Understanding the functional role of these proteins in their organismal and environmental contexts remains a major challenge because most of the organisms that house such proteins are difficult to culture and laboratory validation is often low-

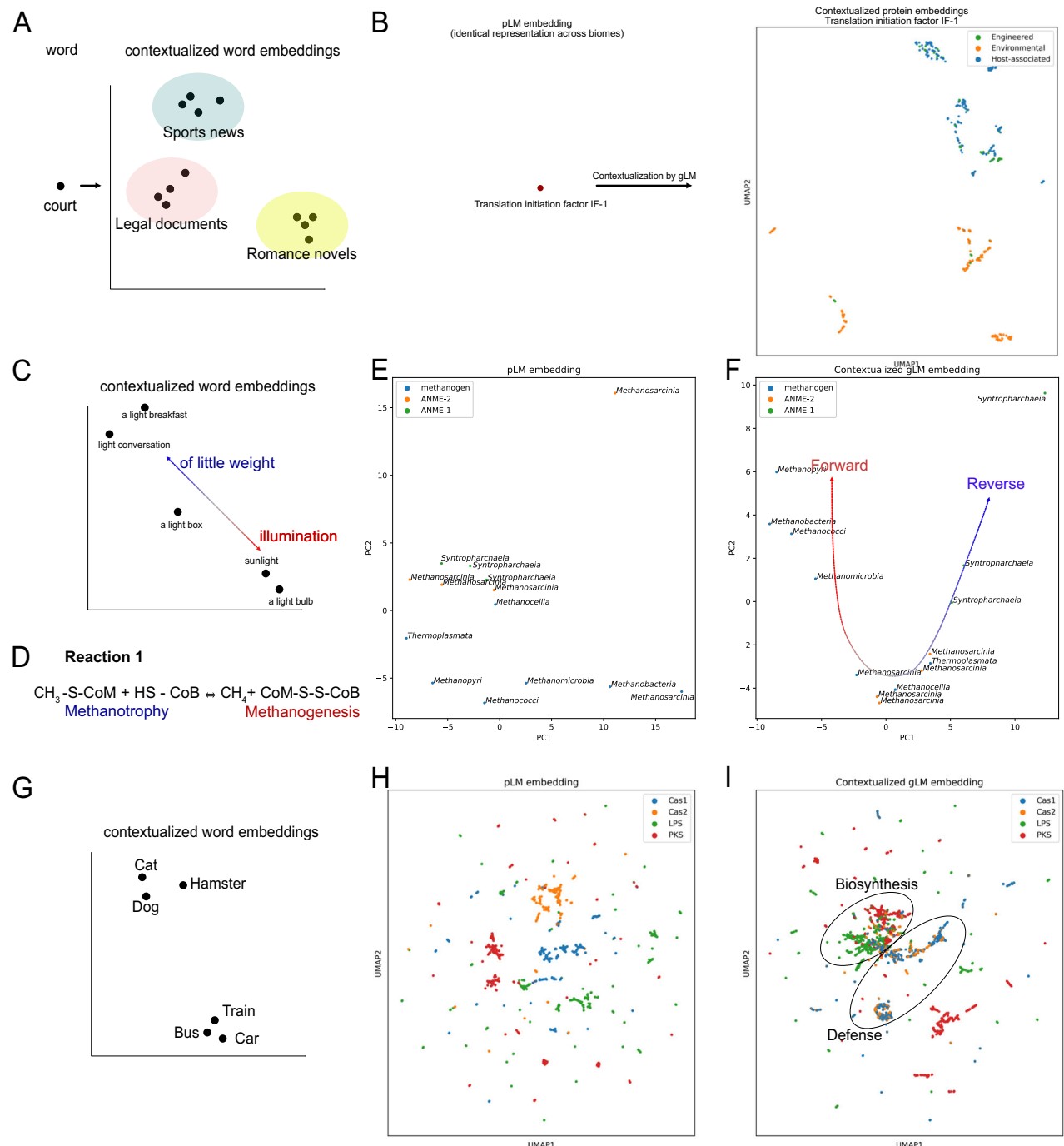

**Fig. 2 | Contextualized protein embedding analysis and comparison with concepts in natural language modeling. A** A word upon contextualization can be mapped to embedding space. For many words, the semantic meaning varies in different types of literature, and therefore their contextualized embeddings cluster with source text type. Figure was created for qualitative visualization. **B** The input protein embedding (output of ESM2 and context-free protein embedding) is the same across all occurrences of the protein in the database. Upon contextualization with gLM, contextualized protein embeddings of the same protein (last hidden layer of gLM at inference time) cluster with biome type, analogous to the source text type in natural language (**A**). Contextualization of 30 other multi-biome MGYPs can be found in Supplementary Fig. 3. **C** A word's meaning upon contextualization varies across a continuous spectrum and can be ambiguous even with contextualization (e.g. double entendre). **D** Reaction 1, carried out by the MCR complex, either backward (Methanotrophy) or forward (Methanogenesis). **E** Principal

Component Analysis (PCA) of context-free protein embeddings of McrA sequences in genomes (total explained variances = 0.56), colored by metabolic classification of the organism (ANME, methanogen) based on previous studies and labeled by class-level taxonomy. **F** PCA of contextualized McrA embeddings (total explained variance = 0.68), where gLM embeddings cluster with the direction of Reaction 1 that the MCR complex is likely to carry out. **G** Geometric relationship between contextualized protein embeddings based on the semantic closeness of words. **H** Input (context-free) protein embeddings of Cas1, Cas2, lipopolysaccharide synthases (LPS) and polyketide synthases (PKS) showing clustering based on structural and sequence similarity. **I** Clustering of contextualized protein embeddings where phage defense proteins cluster (Cas1 and Cas2) and biosynthetic gene products cluster (lipopolysaccharide synthases [LPS] and polyketide synthases [PKS]). Source data are provided as a Source Data file.

throughput. In microbial genomes, proteins conferring similar functions are found in similar genomic contexts due to selective pressures bestowed by functional relationships (e.g. protein-protein interactions, co-regulations) between genes. Based on this observation, we posited that contextualization would provide richer information that pushes the distribution of unannotated genes closer to the distribution of annotated genes. We compared the distributions of unannotated and annotated fractions of proteins in our dataset using context-free pLM embeddings and contextualized gLM embeddings. We found a statistically significant lower divergence between distributions of unannotated and annotated genes in gLM embeddings compared to pLM embeddings (paired t-test of Kullback-Leibler divergences, t-test statistic = 7.61, two-sided, p-value < 1e-4, $n = 10$; see Methods for sampling and metric calculation). This suggests a greater potential for using gLM embeddings to transfer validated knowledge in cultivable and well-studied strains (e.g. *E. coli* K-12) to the vastly uncultivated metagenomic sequence space. Genomic context, along with molecular structure and phylogeny, appear to be important information to abstract in order to effectively represent sequences such that we can uncover hidden associations between the known and the unknown fractions of biology.

## Contextualization improves enzyme function prediction

To test the hypothesis that the genomic context of proteins can be used to aid function prediction, we evaluated how contextualization can improve the expressiveness of protein representations for enzyme function prediction. First, we generated a custom MGYP-EC dataset where the train and test data were split at 30% sequence identity for each EC class (see Methods). Second, we apply a linear probe (LP) to compare the expressiveness of representations at each gLM layer, with and without masking the queried protein (Supplementary Fig. 5). By masking the queried protein, we can assess gLM's ability to learn functional information of a given protein, only from its genomic context, without the propagation of information from the protein's pLM embeddings. We observed that a large fraction of contextual information pertaining to enzymatic function is learned in the first six layers of gLM. We also demonstrate that context information alone can be predictive of protein function, reaching up to 24.4 ± 0.8% accuracy. In contrast, without masking, gLM can incorporate information present in the context with the original pLM information for each queried protein. We observed an increase in expressivity of gLM embeddings also in the shallower layers, with accuracy reaching up to 51.6 ± 0.5% in the first hidden layer. This marks a 4.6 ± 0.5% increase from context-free pLM prediction accuracy (Fig. 3A) and 5.5 ± 1.0% increase in mean average precision (Fig. 3C) Thus, we demonstrate that information that gLM learns from the context is orthogonal to information captured in pLM embedding. We also observed diminishing expressivity in enzyme function information with deeper layers of gLM; this is consistent with previous examinations of LLMs, where deeper layers are specialized to the pretraining task (masked token prediction), and is consistent with previous examinations of LLMs, where the best-performing layer depends on the specific downstream tasks[39]. Finally, to further examine the expressiveness of these representations, we compared per-class F1 score gains (Fig. 3B). We observe statistically significant differences in F1 scores (t-test, two-sided, Benjamini/Hochberg corrected *p* value < 0.05, $n = 5$) between the two models in 36 out of 73 EC classes with more than ten samples in the test set. Majority (27 out of 36) of the statistical differences resulted in improved F1 score in LP trained on gLM representations.

## Horizontal transfer frequency corresponds to genomic context embedding variance

A key process that shapes microbial genome organization and evolution is horizontal gene transfer (HGT). The taxonomic range in which genes are distributed across the tree of life depends on their function

and the selective advantage they incur in different environments. Relatively little is known about the specificity in the genomic region into which a gene gets transferred across phylogenetic distances. We examined the variance of gLM embeddings for proteins that occur at least one hundred times in the database. Variance of gLM-learned genomic contexts are calculated by taking a random sample of 100 occurrences and then calculating the mean pairwise distances between the hundred gLM embeddings. We conduct such independent random sampling and distance calculation ten times per gene and then calculate the mean value. As a baseline, we calculate variance of subcontig-averaged pLM embeddings using the same sampling method, to compare the information learned from training gLM. Our results show that gLM-learned genomic context variances have a longer right-hand tail (kurtosis = 1.02, skew = 1.08) compared to the contig-averaged pLM baseline that is more peaked (kurtosis = 2.2, skew = 1.05) (Fig. 3D). Notably, the most context-variant genes in the right tail of gLM-learned context variance distribution (orange) included phage genes and transposases, reflecting their ability to self-mobilize. Interestingly, we did not find any phage genes in the right-most tail of contig-averaged pLM embedding variance distribution (blue), although we did find genes involved in transposition (Supplementary Table 4). gLM-learned genomic context variances can be used as a proxy for horizontal transfer frequencies and can be used to compare the fitness effects of the genomic context on the evolutionary trajectory (e.g. gene flow) of genes, as well as to identify undercharacterized and functional transposable elements.

## Transformer's attention captures operons

The transformer attention mechanism[24] models pairwise interaction between different tokens in the input sequence. Previous examinations of the attention patterns of transformer models in natural language processing (NLP)[39] have suggested that different heads appear to specialize in syntactic functions. Subsequently, different attention heads in pLMs[40] have been shown to correlate to specific structural elements and functional sites in a protein. For our gLM, we hypothesized that specific attention heads focus on learning operons, a "syntactic" feature pronounced in microbial genomes where multiple genes of related function are expressed as single polycistronic transcripts. Operons are prevalent in bacterial, archaeal and their viral genomes, while rare in eukaryotic genomes. We used the *E.coli* K-12 operon database[41] consisting of 817 operons for validation. gLM contains 190 attention heads across 19 layers. We found that heads in shallower layers correlated more with operons (Fig. 4A, Supplementary Fig. 6, with raw attention scores in the 7th head of the 2nd layer [L2-H7] linearly correlating with operons with 0.44 correlation coefficient (Pearson's rho, Bonferroni adjusted *p* value < 1E-5) (Fig. 4B). We further trained a logistic regression classifier (operon predictor) using all attention patterns across all heads. Our classifier predicts the presence of an operonic relationship between a pair of neighboring proteins in a sequence with high precision (mean average precision = 0.775 ± 0.028, five-fold cross-validation) (Fig. 4C). We baseline this performance by training an operon predictor on the one-hot amino acid representation-based gLM ablation (mean average precision = 0.426 ± 0.015, five-fold cross-validation; Supplementary Table 3), that learns from the orientation and co-occurrence information but cannot fully leverage rich representation of genes.

## Context dependency of AAA+ regulator functions in complex genetic systems

Understanding the functional role of a regulatory protein in an organism remains a challenging task because the same protein fold may carry out different functions depending on the context. For instance, AAA+ proteins (ATPases associated with diverse cellular activities) utilize the chemical energy from ATP hydrolysis to confer diverse mechanical cellular functions[42]. However, AAA+ regulators can

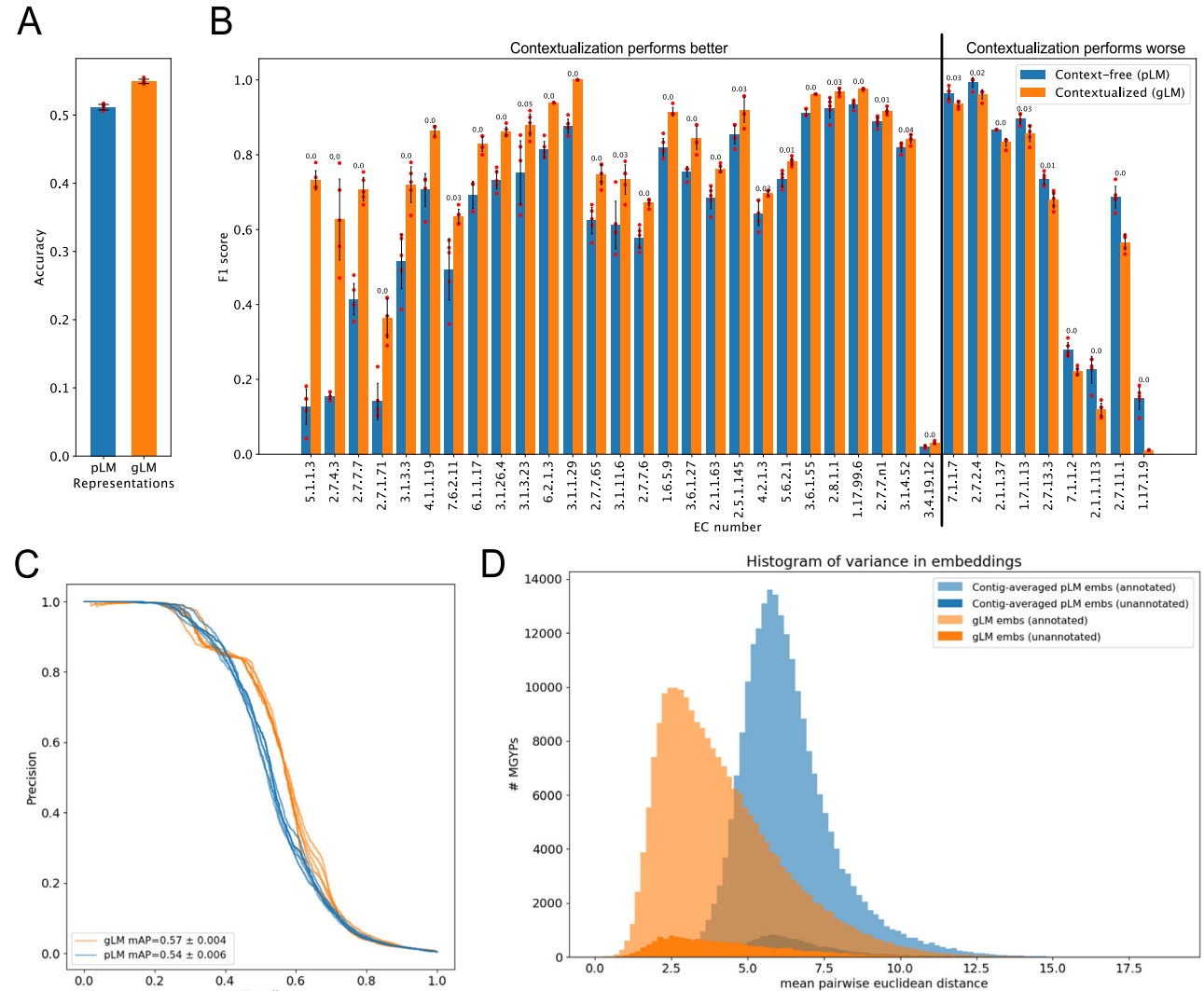

**Fig. 3 | Contextualization of gene function. A** Linear probe enzyme commission (EC) number classification accuracy for pLM (ESM2) representations and gLM (1st hidden layer) representations. Data are presented as mean values +/- standard deviation over five technical replicates. **B** F1-score comparisons of statistically significant (t-test, two-sided, Benjamini/Hochberg corrected *p* value < 0.05, technical replicates = 5) differences in performance of pLM- and gLM-based EC number linear probes. EC classes are ordered with the largest gain with contextualization on the left to the largest loss with contextualization on the right. Data are presented as mean values +/- standard deviation. Adjusted p-value (with two significant figures) for each class is specified above the bars. **C** Precision-Recall curves of pLM- and gLM-based EC number linear probes. **D** Histogram of variance (# bins = 100) calculated using contextualized embeddings (gLM; orange) and contig-averaged pLM (blue) embeddings of MGYPs that occur at least 100 times in the database. Histograms for unannotated and annotated fraction of the MGYPs are plotted separately and bars are not stacked. Annotated examples in the long right tail include phage proteins and transposases, reflecting their ability to self-mobilize (see annotations of top tens most variant genes in Supplementary Table 4). Source data are provided as a Source Data file.

also play very different, broad functional roles depending on their cellular interacting partners from protein degradation and DNA replication to DNA transposition. One particularly interesting example is the TnsC protein, which regulates DNA insertion activity[43] in Tn7-like transposon systems. Multiple bioinformatic efforts focused on discovery of previously uncharacterized transposons through metagenome search[44] and sequence searches of assembled genomes[45] aimed at identifying suitable homologs for genome-editing applications[46]. In order to test whether the methods developed here could identify Tn7-like transposition systems as well as distinguish these from other functional contexts, we explored the contextualized semantics of TnsC's structural homologs in the MGnify database. Without contextualization, there appears no clustering with associated transposase activity (KL divergence ratio = 1.03; see Methods for calculation of this metric, Fig. 4E). However, with added contextualization, previously identified TnsC (orange) and manually inspected TnsC-like structural homolog (red, labeled "TnsC-like") cluster together (KL divergence

ratio = 0.38; Fig. 4F; see Supplementary Fig. 7B, C for comparison with gLM-only and contig-averaged pLM baselines). We further validate this visualization using embedding distance-based clustering (Supplementary Fig. 8). Many structural homologs of TnsC were not involved in transposition and this is reflected in distinct clusters of gray data points away from known TnsC (oranges) and TnsC-like structural homologs (red) in Fig. 4F. These clusters represent diverse and context-dependent AAA+ regulation activity that cannot be predicted from neither structure nor raw sequence alone. We predicted an operonic relationship between these AAA+ regulators and their neighboring genes and found many to be in operonic relationships with gene modules of diverse function, including pilus assembly and viral host-nuclease inhibition (Fig. 4D, Supplementary Fig. 7A). In some cases, queried AAA+ proteins did not appear to be in an operonic association with the neighboring proteins, suggesting some AAA+ proteins are less likely to be functionally associated with their neighbors than others (Supplementary Fig. 7A, example 6). Using this

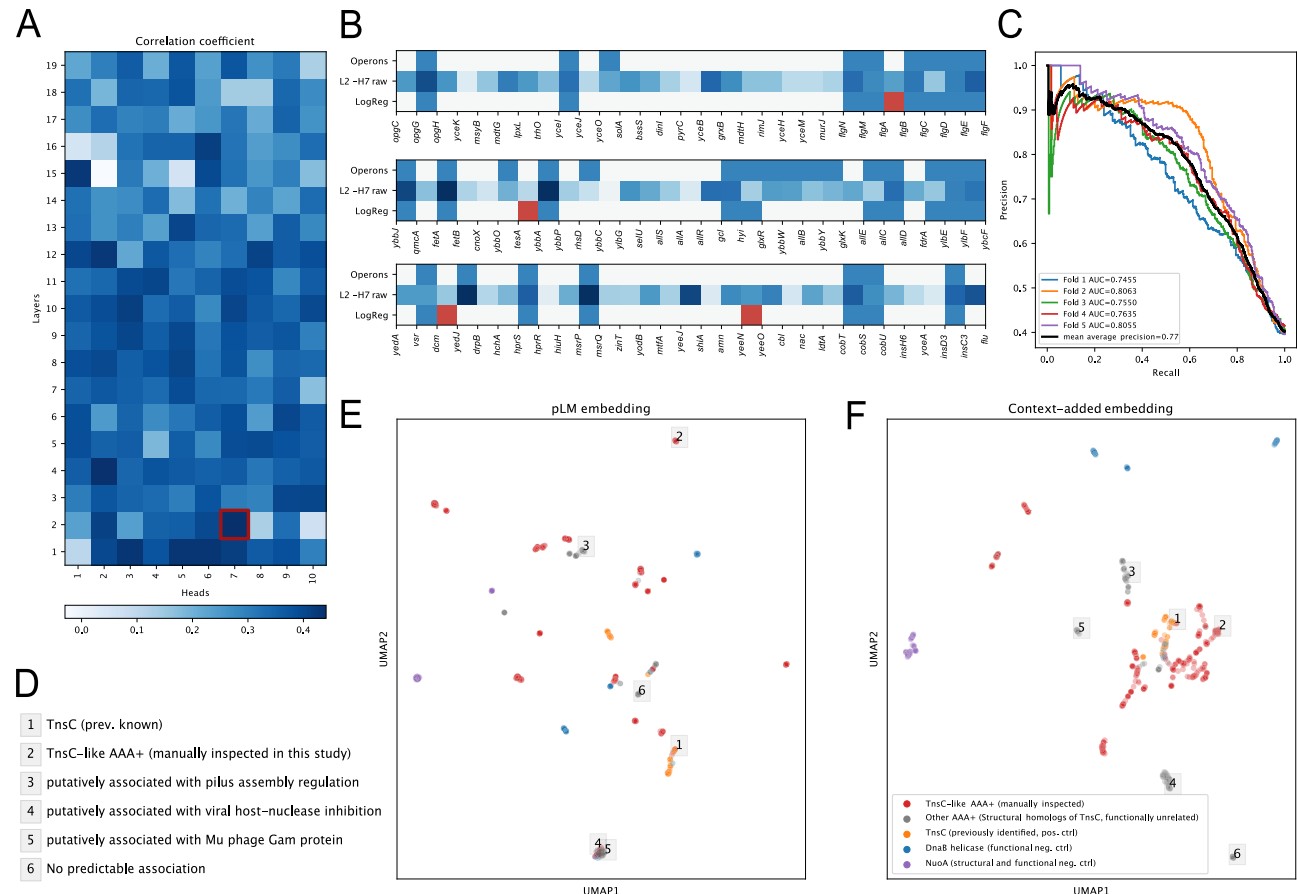

**Fig. 4 | Attention analysis. A** Correlation coefficients (Pearson's rho) between attention heads across layers and operons. Darker color corresponds to stronger correlation with previously identified operons. Attention patterns of the second layer-seventh head [L2-H7] is most strongly correlated with the operons. **B** Three random examples of contigs and predicted operonic relationship between neighboring proteins. Proteins are listed in the order they are encoded in the contig. Ground truth *E.coli* K-12 operons (top row), raw attention scores in the attention head [L2-H7] most correlated with operons (middle row) and logistic regression prediction using all attention heads (last row) where false positive predictions (or possibly misannotated ground truths in the case of flagellar proteins in the first example) are marked in red. **C** Five-fold cross-validation precision-recall curves of logistic regression trained using all operons and attention heads. **D** AAA+ regulator associations characterized using attention-based prediction of operons (**Extended Fig. 11A**) corresponding to labeled examples in panels **E** and **F**. **E** ESM2 generated input protein embeddings of AAA+ regulator proteins that are structural homologs

to TnsC (grey and red; using Foldseek[60]). Structural homologs of TnsC with confirmed involvement in Tn7-like transposons upon manual inspection were designated "TnsC-like AAA+ (manually inspected)" and are colored red. Other MGYP proteins annotated as "TnsC" against the UniRef90 database (orange) were added as positive controls for TnsC function. NuoA (NADH-quinone oxidoreductase subunit A; purple) were added as structural and functional negative controls. DnaB helicases (blues) were added as functional negative controls, as these proteins have similar folds to TnsC but are not associated with transposition. **F** Combined input protein and context embeddings of genes in panel **E**. These embeddings are generated through concatenation of pLM (ESM2) embeddings and context (last layer of gLM) embeddings. Negative controls (NuoA and DnaB helicases) form distinct clusters in both **E** and **F**. Numbered labels in grey boxes indicate the AAA+ proteins with various functional association predictions listed in panel **D** and Supplementary Fig. 7. Raw distance based clustering of the embeddings are shown in Supplementary Fig. 8. Source data are provided as a Source Data file.

example of AAA+ regulators, we illustrate that combining the contextualized protein embeddings and attention-based operon interaction may provide an important avenue for exploring and characterizing the functional diversity of regulatory proteins.

## gLM predicts paralogy in protein-protein interactions

Proteins in an organism are found in complexes and interact physically with each other. Recent advances in protein-protein interaction (PPI) prediction and structural complex research has largely been guided by identifying interologs (conserved PPI across organisms) and co-evolutionary signals between residues[47]. However, distinguishing paralogs from orthologs (otherwise known as the "Paralog matching" problem) in the expanding sequence dataset remains a computational challenge requiring queries across the entire database and/or phylogenetic profiling. In cases where multiple interacting pairs are found within an organism (e.g. histidine kinases (HK) and response regulators (RR)), prediction of interacting pairs is particularly difficult[48]. We

reasoned that gLM, although not directly trained for this task, may have learned the relationships between paralogs versus orthologs. In order to test this capability, we used a well-studied example of interacting paralogs (ModC and ModA, Fig. 5A) which form an ABC transporter complex. We queried gLM to predict the embedding of an interacting pair given no context except the protein sequence of either ModA or ModC. We find that without any fine-tuning gLM performs at least an order of magnitude better than what is expected by random chance (see Methods). Specifically, for 398 out of 2700 interacting pairs, gLM makes predictions that belong to the same cluster (50% sequence identity, $n = 2100$ clusters) as the true label, and in 73 pairs, the gLM predicts a label that is closest to the exact interacting pair (simulated random chance expected match=$1.6 \pm 1.01$, $n = 10$) (Fig. 5B). Importantly, when considering only very high confidence predictions (prediction likelihood > 0.9, $n = 466$), gLM is able to match paralogs with an increased accuracy of 25.1%. When paralogs are correctly paired, gLM is more confident about the prediction (average

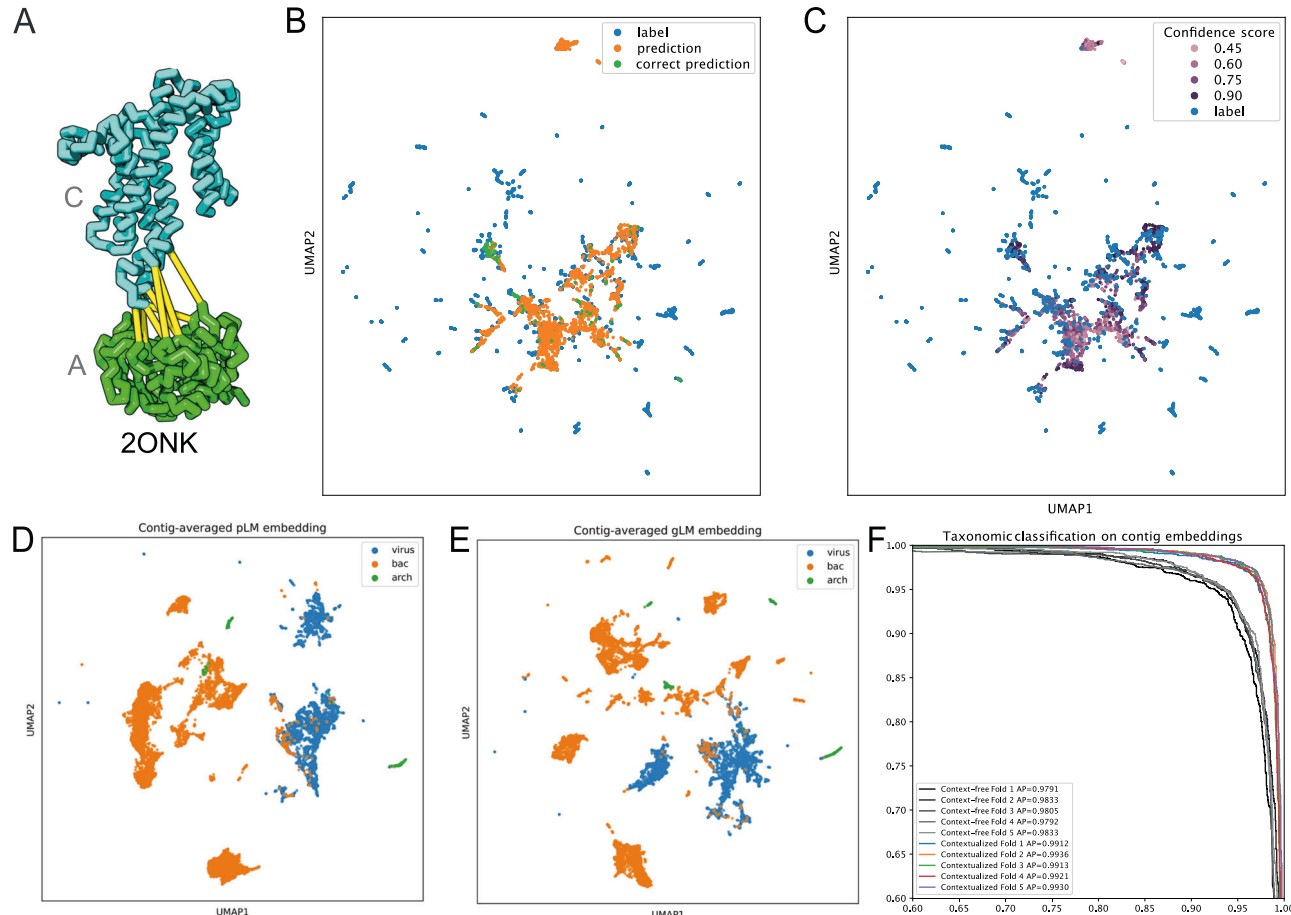

**Fig. 5 | Potential for transfer learning. A** ModA and ModC interaction (protein data bank structure 2ONK)[47] **B** UMAP projection of predictions (orange) and labels (blues) of paralogs (ModAC shown in A), where correct predictions are colored in green. **C** Predicted embeddings are colored based on the predicted confidence. Out of distribution predictions and predictions closer to the mean are generally of lower confidence, while correct predictions are of higher confidence. **D, E** Random 30-gene contigs from representative bacterial ("bac") and archaeal ("arch") genomes and reference viral ("vir") genomes were embedded by mean-pooling ESM2 protein embeddings (context-free contig embeddings, **D**) and by mean-pooling the last hidden layer of gLM (contextualized contig embeddings, **E**). **F** Micro-averaged precision-recall curves and average precisions for logistic regression classifiers trained using context-free contig embeddings (grey lines) and contextualized contig embeddings (colored lines) for class-level taxonomy classification task. Each line represents a fold in stratified k-fold cross-validation (k = 5). Class-level taxonomy for each contig is shown in Supplementary Fig. 9A, B and the confusion matrices for logistic regression classifiers are shown in Supplementary Fig. 9C, D. Source data are provided as a Source Data file.

confidence for correct prediction = 0.79, average confidence across all predictions = 0.53), while less certain predictions are either out of distribution, or closer to the mean of labels (Fig. 5C). We attribute part of the inaccuracies in prediction due to the fact that gLM was not trained on the task of predicting a masked gene given only a single gene as genomic context, though we expect the performance to improve with expanding the training sequence length range and fine-tuning the model specifically for the "paralog matching" problem.

**Contextualized contig embeddings and potential for transfer learning**

Contextualized protein embeddings encode the relationship between a specific protein and its genomic context, retaining the sequential information within a contig. We hypothesized that this contextualization adds biologically meaningful information that can be utilized for further characterization of the multi-gene genomic contigs. Here, we define a contextualized contig embedding as a mean-pooled hidden layer across all proteins in the subcontig, and a context-free contig embedding as mean-pooled ESM2 protein embeddings across the sequence (see methods). Both embeddings consist of 1280 features. We test our hypothesis by examining each of these embeddings' ability to linearly distinguish viral sequences from bacterial and archaeal

subcontigs. In metagenomic datasets, the taxonomic identity of assembled sequences must be inferred post-hoc, therefore the identification of viral sequences is conducted based on the presence of viral genes and viral genomic signatures[49]. However, such classification task remains a challenge particularly for smaller contig fragments and less characterized viral sequences. Here, we sampled random 30-protein subcontigs from the representative bacterial and archaeal genome database and reference viral genomes in the NCBI and visualized their context-free contig embeddings (Fig. 5D) and contextualized contig embeddings (Fig. 5E). We observed more separation and taxonomic clusters at both domain- and class-levels (Supplementary Fig. 9), suggesting that taxonomic signature is enhanced by encoding the latent relationships between proteins. This is further validated by training a logistic regression classifier on context-free and contextualized contig embeddings for class-level taxonomy (Supplementary Fig. 9A, B), where we see a statistically significant improvement in average precision (Fig. 5F, see Supplementary Fig. 7C, D for confusion matrices). This emphasizes the biological importance of a protein's relative position in the genome and its relationship with the genomic context, and further indicates that this information can be effectively encoded using gLM. Contextualized contig embeddings present opportunities for transfer learning beyond viral sequence

prediction, such as improved metagenomically-assembled genome (MAG) binning and assembly correction.

## Discussion

The unprecedented amount and diversity of metagenomic data, coupled with advances in deep learning, presents exciting opportunities for building models that can learn hidden patterns and structures of biological systems. Such models build upon the conceptual and statistical frameworks that evolutionary biologists have developed for the past century. With capabilities of abstracting much larger amounts of data, these models can disentangle the extraordinary complexity of organismal genomes and their encoded functions; this is a key step in furthering our understanding of biological processes. The work presented here demonstrates and validates the concept of genomic language modeling. Our implementation of the masked genomic language modeling illustrates the feasibility of training such a model, and provides evidence that biologically meaningful information is being captured in learned contextualized embeddings and yielding meaningful interpretations of the attention patterns. We show that gLM can be used for diverse downstream tasks, including enzyme function prediction, operon prediction, paralog matching and contig taxonomy prediction. Furthermore, we demonstrate gLM's ability to illuminate context dependency in functions across structural and sequence homology through the example of AAA+ regulators. Taken together, gLM presents a highly promising direction for interpreting biology and we propose key areas for further development: First, the transformer architecture has shown to be successful in efficient scaling; in both natural language[50] and protein language processing[23], increasing the number of parameters in the model along with the training dataset size have been shown to lead to vastly improved performance and generalizability. Our model consists of ~1B parameters which is at least a magnitude smaller compared to state-of-the-art pLMs. With further hyperparameter tuning and scaling, we expect better performance of the model. Second, our model currently uses pLM embeddings to represent proteins in the input. These embeddings are generated by mean-pooling the amino acid residue-level hidden states across the protein sequence, and therefore the residue-specific information and synonymous mutation effects are likely obscured. Future iterations of the model could use raw residue-level or codon-level embeddings as input to allow modeling of residue-to-residue co-evolutionary interactions between proteins and synonymous mutation effects on gene function. Third, the task of reconstructing masked protein embeddings requires modeling a distribution over possible embeddings; our method approximates this distribution using a fixed number of predictions. Future work could improve upon this by using a generative approach, such as a diffusion or GAN model. This may allow for better prediction accuracy and greater generalizability for unseen datasets. Fourth, adding non-protein modalities (e.g. non-coding regulatory elements) as input to gLM may also greatly improve gLM's representation of biological sequence data, and can learn protein function and regulation conditioned upon other modalities[51]. Finally, our model was trained largely on bacterial, archaeal and viral genomes, therefore, how this method can be adapted for eukaryotic genomes, especially those with extensive intergenic regions, remains to be further explored.

One of the most powerful aspects of the transformer-based language models is their potential for transfer learning and fine-tuning. We tested some of the capabilities of gLM and successfully showed that higher-order biological information, including gene function and regulation can be learned using genomic sequences. Our results highlight the importance of contextualization of biological data, particularly as we scale our modeling efforts from biomolecules to whole organisms. We propose the following promising future directions for applying gLM for advancing biological research. 1) Feature-based transfer learning for predicting protein function (e.g. Gene Ontology [GO] term), particularly those with limited sequence and structural homology. 2) Fine-tuning gLM for the protein-protein-interactome prediction task. 3) Using gLM features to encode genomic contexts as additional input for improved and contextualized protein structure predictions. In conclusion, genomic language modeling is a powerful tool to unbiasedly condense important biological information from full metagenomic sequences. Coupled with the advances in long-read sequencing, we expect a drastic increase in the input data quality, quantity and diversity. Genomic language modeling presents an avenue to bridge the gap between atomic structure and organismal function, and thereby brings us closer to modeling biological systems, and ultimately, manipulating biology with precision (e.g. genome editing, synthetic biology).

## Methods

### Sequence database

The genomic corpus was generated using the MGnify[27] dataset (released 2022-05-06 and downloaded 2022-06-07). First, genomic contigs with greater than 30 genes were divided into 30 gene non-overlapping subcontigs resulting in a total of 7,324,684 subcontigs with lengths between 15 and 30 genes (subcontigs <15 genes in length were removed from the dataset). We chose 30 as maximum context length because while longer context results in higher modeling performance (Supplementary Fig. 10A), 67% of the raw MGnify contigs with > 15 genes were of =<30 genes in length (Supplementary Fig. 10B), and therefore increasing the context length beyond 30 would have resulted in many examples with padding (reduced computational efficiency). Each gene in the subcontig was mapped to a representative protein sequence (representative MGYP) using mmseqs/linclust[52], with coverage and sequence identity thresholds set at 90% (pre-computed in the MGnify database), resulting in a total of 30,800,563 representative MGYPs. Each representative MGYP was represented by a 1280-feature protein embedding, generated by mean-pooling the last hidden layer of the ESM2[23] "esm2_t33_650M_UR50D" model. Due to the memory limitation in computing embeddings for very long sequences, 116 of the MGYP sequences longer than 12290 amino acids were truncated to 12290 amino acids. ESM2 embeddings were normalized (by subtracting the mean of each feature and dividing by its standard deviation) and clipped such that all features range from −10 to 10, to improve training stability. A small fraction (0.4%) of the genes could not be mapped to a representative MGYP and therefore the corresponding sequence information could not be retrieved from the MGnify server; these sequences were assigned a 1280 feature vector of ones. For each gene in the sub-sequence, we added a gene orientation feature to the standardized MGYP protein embedding, where 0.5 denotes "forward" orientation relative to the direction of sequencing, and −0.5 denotes "reverse" orientation. Thus, each gene was represented by a 1281 feature vector in our corpus.

### gLM architecture and training

gLM was built on the huggingface implementation of the RoBERTa[53] transformer architecture. gLM consisted of 19 layers with hidden size 1280 and ten attention heads per layer, with relative position embedding ("relative_key_query")[54]. For training, 15% of the tokens (genes) in the sequence (subcontig) were randomly masked to a value of −1. We then tasked the model with the objective of predicting the label of the masked token, where the label consists of a 100-feature vector that consists of the PCA whitened 99 principal components (explained variance = 89.7%. Supplementary Fig. 11) of the corresponding ESM2 protein embedding concatenated with its orientation feature. Reduced dimensionality of labels using PCA increased the stability of training. Specifically, gLM projects the last hidden state of the model into four 100-feature vectors and four corresponding likelihood values using a

linear layer. Total loss is calculated using the following Eq. (1).

$$MSE(\text{closest prediction,label}) + \alpha * CrossEntropyLoss(\text{likelihoods,closest prediction index}) \tag{1}$$

The closest prediction is defined as the prediction that is closest to the label, computed by L2 distance. We set $\alpha = 1e-4$. gLM was trained in half-precision with batch size 3000 with distributed data parallelization on four NVIDIA A100 GPUs over 1,296,960 steps (560 epochs), including 5000 warm-up steps to reach a learning rate of 1e-4 with AdamW[55] optimizer.

### Performance metric and validation
In order to evaluate the model quality and its generalizability beyond the training dataset, we use a pseudo-accuracy metric, where we deem a prediction to be "correct" if it is closest in Euclidean distance to the label of the masked gene relative to the other genes in the subcontig. Pseudo-accuracy calculation is described in Eq. (2).

$$\text{pseudo accuracy} = \frac{\#count(argmin(dist(\text{prediction,labels in subcontig})) == index(\text{masked gene}))}{\#\text{masked genes}} \tag{2}$$

We chose to validate our metric and subsequent analyses on the best-annotated genome to date: *E.coli* K-12[56]. In order to remove as many *E.coli* K-12 like subcontigs from the training dataset, we removed 5.2% of the subcontigs in which more than half of the genes were > 70% similar (calculated using mmseqs2 search[52]) in amino acid sequence to *E.coli* K-12 genes. We validate our pseudo accuracy metric by calculating the absolute accuracy on the *E.coli* K-12 genome for which each gene was masked sequentially (Eq. (3))

$$\text{absolute accuracy} = \frac{\#count(argmin(dist(\text{prediction,all genes in E.coli K}-12)) == index(\text{masked gene}))}{\#\text{genes in E.coli K}-12} \tag{3}$$

### Contextualized embedding calculation and visualization
Contextualized protein embedding of a gene is calculated by first inputting a 15-30 gene subcontig containing the gene of interest, and then running inference on the subcontig using the trained gLM without masking. We then use the last hidden layer of the model corresponding to the gene as the embedding consisting of 1280 features.

### Gene annotation
Genes were annotated using Diamond v2.0.7.145[57] against the UniRef90 database[58] with an e-value cut-off 1E-5. Genes were labeled as "unannotated" if either 1) no match was found in the UniRef90 database, or 2) the match was annotated with following keywords: "unannotated", "uncharacterized", "hypothetical", "DUF"(domain of unknown function).

### McrA protein analysis
McrA protein encoding Methanogens and ANME genomes were selected from the accession ID list found in the supplement of Shao et al.[35]. subcontigs containing *mcrA* were extracted with at most 15 genes before and after *mcrA*. The context-free and contextualized embeddings of McrA were calculated using the ESM2 and gLM, respectively.

### Distributions of unannotated and annotated embeddings
Distributions of unannotated and annotated embeddings in the database were compared using Kullback-Leibler (KL) divergence analysis.

First, ten random samples of 10,000 subcontigs from the MGnify corpus. pLM and gLM embeddings of the genes were calculated using mean-pooled last hidden layer of ESM2 embeddings and mean-pooled last hidden layer of gLM, respectively. Outliers were removed using Mahalanobis distance and a chi-squared threshold of 0.975. pLM and gLM embedding dimensions were reduced to 256 principal components ($91.9 \pm 1.72\%$ and $80.1 \pm 6.89\%$ total variances explained, respectively). KL divergence was calculated using the following Eq. (4).

$$D_{KL}(P||Q) = \frac{1}{2}\left(tr(\Sigma_1^{-1}\Sigma_0) - k + (\mu_1 - \mu_2)^T\Sigma_1^{-1}(\mu_1 - \mu_0) + \ln\left(\frac{\det\Sigma_1}{\det 0}\right)\right) \tag{4}$$

where P corresponds to the distribution of unannotated genes and Q corresponds to the distribution of annotated genes, with $\mu_0,\mu_1$ respectively as means and $\Sigma_0,\Sigma_1$ respectively as covariance matrices. The significance of the KL divergence differences between pLM and gLM embeddings is calculated using a paired t-test across the ten samples.

### Enzyme Commission number prediction
Custom MGYP-Enzyme Commission (MGYP-EC) dataset was created by first searching (mmseqs2[52] with default setting) MGYPs against the "split30.csv" dataset previously used to train CLEAN[59]. "split30.csv" dataset consists of EC numbers assigned to UniProt sequences clustered at 30% identity. Only MGYP hits with >70% sequences to "split30.csv" were considered and MGYPs with multiple hits with >70% similarity were removed. Test split was selected by randomly selecting 10% of "split30.csv" UniProt IDs in each EC category that map to MGYPs. EC categories with less than four distinct UniProt IDs with MGYP mapping were removed from the dataset, resulting in 253 EC categories. The train set consisted of MGnify subcontigs in the corpus that contained at least one the 27936 MGYPs mapping to 1878 UniProt IDs. The test set consisted of randomly selected MGnify subcontig containing each of 4441 MGYPs mapping to 344 UniProt IDs. pLM (context-free) embeddings were calculated for each of MGYP with EC number assignment by mean-pooling the last hidden layer of its ESM2 embedding. Masked (context-only) gLM embeddings were calculated for each of the 19 layers by running inference on subcontigs with masks at the positions of MGYPs with EC number assignment and subsequently extracting per-layer hidden representations for masked positions. gLM (contextualized) embeddings were calculated also for each layer by running inference without masking and subsequently extracting per-layer hidden representations for MGYPs with EC number assignments. Linear probing was conducted for these embeddings with a single linear layer. Linear probes were trained with early stopping (patience = 10, github.com/Bjarten/early-stopping-pytorch/blob/master/pytorchtools.py) and batch size = 5000, and training results were replicated five times with random seeds to calculate error ranges.

### Variance of contextualized protein embedding analysis
Contextualized protein embeddings are generated at inference time. Variances of contextualized protein embeddings were calculated for MGYPs that occur at least 100 times in the dataset, excluding the occurrences at the edges of the subcontig (first or last token). For each such MGYP, we take 10 random independent samples consisting of 100 occurrences and calculate the mean pairwise euclidean distances between the contextualized embeddings. To assess the role gLM plays in contextualization, we used the above sampling method to calculate the variance of contig-averaged pLM embeddings (pLM embeddings mean-pooled across the contig) for each MGYP that occurs at least 100 times in the dataset.

## Attention analysis

Attention heads (n = 190) were extracted by running inference on unmasked subcontigs, and the raw attention weights were subsequently symmetrized. *E.coli* K-12 RegulonDB[56] was used to probe heads with attention patterns that correspond the most with operons. Pearson's correlation between symmetrized raw attentions and operons were calculated for each head. We trained a logistic regression classifier that predicts whether two neighboring genes belong to the same operon based on the attention weights across all attention heads corresponding to the gene pair.

## TnsC structural homolog analysis

TnsC structural homologs were identified by searching ShCAST TnsC (PDB 7M99 chain H) against the MGYP database using Foldseek[60] on ESM Atlas (https://esmatlas.com/). The contigs containing these homologs in the MGnify database were used to calculate the contextualized protein embeddings of the identified structural homologs. Contigs with less than 15 genes were excluded from the analysis. Contigs encoding proteins that were previously identified as "TnsC" using the UniRef90 database (see Gene annotation methods section above) were included in the database. "TnsC-like" contigs were manually annotated based on the presence of transposase genes (TnsB) and TniQ. Fifty random examples of MGnify contigs containing MGYPs annotated as NuoA and DnaB were added as negative controls for the UMAP visualization. We calculated KL divergence ratios using the following Eq. (5).

$$\frac{D_{KL}(B||A)}{D_{KL}(C||A)} \qquad (5)$$

where A is the distribution of representations of known TnsC, B is the distribution of representations of manually curated TnsC-like AAA+ regulators, C is the distribution of representations of other AAA+ regulators that are functionally unrelated structural homologs of known TnsC. Therefore, this metric ranges from 0 to 1, where a lower ratio represents increased ability to functionally discriminate distribution of B from C relative to A. KL divergence was calculated using the same formula as in the methods section Distributions of unannotated and annotated embeddings, except with 20 principal components that explained >85% of variances across all embeddings.

## Paralogy and orthology analysis

UniProt IDs from ABC transporter ModA and ModC protein interacting paralog pairs (*n* = 4823) were previously identified by Ovchinnikov et al.[47] and were downloaded from https://gremlin.bakerlab.org/cplx.php?uni_a=2ONK_A&uni_b=2ONK_C and subsequently used to download raw protein sequences from the UniProt server. Only pairs (*n* = 2700) where both raw sequences were available for download, and where the UniProt ID differed by one (indicating adjacent positioning in the reference genome) were selected for subsequent analyses. We constructed test contigs consisting of three genes, where first and third genes are masked, and the second gene encodes one of the pair in forward direction. We then queried gLM to predict the two neighboring masked genes, and considered the prediction to be correct if either of the proteins closest to masked genes's highest confidence prediction in embedding space belongs to the same sequence cluster as the interacting protein (50% amino acid sequence identity, calculated using CD-HIT v4.6[61]). Random chance correct prediction rate (1.6 ± 1.0 was simulated using 1000 iterations of random predictions generated within the standard normal distribution and performing the same operation as above to compute the rate of correct predictions[62].

## Taxonomic analysis and visualization

4551 bacterial and archeal representative genomes and 11660 reference viral genomes were downloaded from the RefSeq database (ftp://ftp.ncbi.nlm.nih.gov/genomes/refseq) on 12 Feb 2023. A random 30-gene subcontig is chosen and encoded using ESM2, which then were subsequently concatenated with an orientation vector and then used as input for the trained gLM. The last hidden layer was mean-pooled across the sequence to retrieve 1280-feature contextualized contig embeddings. The ESM2 protein embeddings were also mean-pooled across the sequence to retrieve 1280-feature context-free contig embeddings. We trained a logistic regression classifier to predict the class-level taxonomy of subcontigs and evaluated the performance using stratified k-fold cross-validation (k = 5).

## UMAP visualization and statistical tests

All UMAP dimensionality reductions calculated with following parameters: n_neighbors = 15, min_dist = 0.1. Silhouette scores were calculated using the *sklearn* package using the default setting with euclidean distance metric.

## Reporting summary

Further information on research design is available in the Nature Portfolio Reporting Summary linked to this article.

## Data availability

Dataset used for training is available for download from the MGnify server (http://ftp.ebi.ac.uk/pub/databases/metagenomics/peptide_database/2022_05/). The model is available at zenodo under accession number 10.5281/zenodo.7855545. Source data for the main Fig. (2c,e,f,i&h; 3a,b,c&d; 4a,b,c,e&f; 5b,c,d,e&f) and supplementary Figs. (2,3,4,5,7,8,9,10 &11) are provided with this paper as a zip file. Source data are provided with this paper.

## Code availability

Training and inference code and analysis scripts are available at https://github.com/y-hwang/gLM (https://doi.org/10.5281/zenodo.10512240).

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

## Acknowledgements

We would like to thank the EBI MGnify team for generating and maintaining the metagenome database. We would also like to thank Meta AI's ESM developers who made both the folded MGnify proteins structures and source-code openly available. We also thank Simon Roux and Landen Goszashti for insightful discussions. This work was supported by the Gordon and Betty Moore Foundation grant #9208 to P.R.G., NSF OCE-1635365 to P.R.G, and by the National Aeronautics and Space Administration under grant no. 80NSSC18K1140 and 80NSSC19K1427 issued through the NASA Network for Life Detection program to P.R.G. S.O. was supported by NIH Grant No. DP5OD026389 and NSF Grant No. MCB2032259. The computations in this paper were run on the FASRC Cannon cluster supported by the FAS Division of Science Research Computing Group at Harvard University.

## Author contributions

Y.H. prepared the datasets and trained the model with support from A.L.C. and S.O.; E.H.K. and Y.H. conducted the TnsC analysis; A.L.C., S.O. and P.R.G. provided input in analysis and data interpretation; Y.H. wrote the manuscript with input from all authors; All authors read and approved the final manuscript.

## Competing interests

A provisional patent (App. Serial No.: 63/491,019) on this work was filed by Harvard University with YH and SO as inventors. The remaining authors declare no competing interests.
