## [Peer Review File · Nature Communications]

Genomic language model predicts protein co-regulation and functionREVIEWER COMMENTS

Reviewer #1 (Remarks to the Author):

Hwang's paper "Genomic language model predicts protein co-regulation and function" develops a genomic language model (gML) to predict protein function based on operon structure. The model is impressive and well thought out. I only have a few minor comments that need to be addressed:

1) The operon framework that the gLM relies on only works with prokaryotes and their viruses. The authors need to openly address this.

2) The authors miss key contributors to the context-based ML field including PMID: 31520072.

Reviewer #2 (Remarks to the Author):

Hwang et al. present a genomic language model that encodes genes via embeddings of a protein language model that get contextualised by neighbouring genes on metagenomic contigs of 15-30 genes.

The hypothesis of the manuscript is valuable and clear: protein embeddings decontextualised of their genomic environment will miss genomic context important for predictions in gene world. Protein language models encode a demonstrated notion of protein structure and function which can sufficiently represent genes. The authors execute well on the analysis of decontextualised vs. contextualised gene/protein representations with several interesting examples.

Main observations:

- In my opinion, calling the model a "genomic language model" is potentially misleading. Protein language models operate on protein sequences treated as "language" through their building blocks, amino acids. A genomic language model, in my opinion, is a model that operates on nucleotides. The model presented here doesn't explicitly account for nucleotides, making it rather a gene order aware protein language model than a genomic language model, as is also argued in 43-55 of introduction. In my opinion this is important because, for instance, synonymous mutations will be transparent to the model.

- I wonder how this approach stands compared to the likes of Enformer (Avsec et al., 2021), DNABert (Ji et al. 2021), or more recent work like GenSLMs (Zvyagin, 2022), CaLM (Outeiral, 2022) and Nucleotide Transformer (Dalla-Torre et al., 2023), all encoding some notion of "genomic language model", some explicitly trying to address the (long-)context issue (Enformer, GenSLMs & Nucleotide Transformer). Especially interesting in this context is GenSLMs, which includes similar principles to this work: a foundational language modelling piece, yet operating on codons, and a contextualising piece, yet based on diffusion. [Note: conflicting interest on Dalla-Torre and Zvyagin]

- Not a lot of explanation or ablation was given to the choice of PCA as reduction technique for ESM2 representations, and for the "parameter" choices, e.g. PCA up to component 100 (why not 10? Why not 500?). Does 100 include 90% of the variation? What would happen if only 70 or 40% of variation is considered, how does it affect downstream performance?

- I think an important baseline would be a model encoding genes with tokenises, other categorical representations, or simple features like length, nucleotide or codon counts (bag-of-words), etc. At current, it's impossible to assess whether the signal from pLMs influences the downstream predictions.

- Similarly, the authors didn't include an ablation of nucleotide transformer models (mentioned earlier), instead of protein representations. Arguably, these models will have higher sensitivity towards nucleotide changes, which may be the goal of a genomic language model.

- Fig 3 A: I would strongly discourage the use of "cluster distinctly" for these plots, as it's visually impossible to verify, practically questionable (maybe a different UMAP parametrisation would lead to opposite results), and statistically not substantiated.

- Line 202 the authors argue about a 2-3% increase in prediction accuracy. Is this within std. error? Regarding the dataset, my understanding from 513 is that the data is split randomly? Minimally, considering the traditional protein sequence context, train/test should be split according to sequence redundancy. Given that in this case the objective is the generalisation of genomic context, other and/or additional considerations should be included to make sure that the downstream model isn't leveraging leakage of information, e.g. similar genomic arrangement or content between train and test contigs. (Train sample: Gene 1, Gene 2, Gene 3 -> Gene 2 -> EC; Test sample: Gene 3, Gene 2, Gene 1 -> Gene 2 -> EC).

Other:

- Reference 50 may need amendment

- 447-448 "ESM2 embeddings were 448 subsequently standardised and clipped at (-10,10)" -- What's "standardised" or "clipped"?

[Christian Dallago]

Reviewer #3 (Remarks to the Author):

The authors present a new language model that bridges the gap between genomic data and proteins. For this, they use embeddings derived from a pre-trained protein language model (pLM), ESM-2, as input to a newly trained language model (encoder-architecture; (Ro)BERT(a)-like), dubbed gLM (genomic language model). Instead of training either on nucleic or amino acids, the authors mix both modalities by deriving exon-structure from millions of metagenomic scaffolds, represented by per-protein ESM-2 embeddings (average-pooling applied over the length dimension of the protein) which are used to train the proposed genomic language model (gLM) on reconstructing corrupted genes using non-corrupted (genomic) context. Simply put, the authors put per-protein embeddings into their genomic context. After training, the authors benchmark their gLM on a variety of tasks, i.e., enzymatic function, taxonomy, and operon structure. The proposed method has the potential to set the foundation for many novel applications. For example, one can derive embeddings that allow to distinguish proteins with identical sequences based on their genomic context. The paper is well-written with many figures illustrating the major arguments of the authors.

The major problem that I see with the current form of the paper is that a large fraction of its evaluation is based on qualitative analysis of UMAP-projections. Despite being unquestionable interesting, low-dimensional projections s.a. t-SNE or UMAP are susceptible to parameter choice and one should be careful when interpreting the global structures in this space. For example, the analysis of 2E and 2F, or 2H and 2I, or 4E and 4F and 5D and 5F relies mostly on the global structure output by UMAP which can change severely when changing parameters. A more quantitative way could be to compute clustering metrics s.a. silhouette scores for those cases using e.g. Euclidean/Cosine distance

in gLM/pLM embedding space. This allows for a quantitative comparison of the methods and should also allow to quantify which signal is captured by the orthogonal signals captured by pLMs and gLMs, i.e., silhouette score computed for data in 2H and 2I once based on structural aspects and once on functional aspects (phage defense proteins cluster etc).

The same, albeit more complex because of a lack of labels, holds true for e.g. 3A and 3B. Despite the lack of labels, the authors could still compute metrics on transferring annotations from labeled/predicted lookup sets to labeled/predicted query proteins using distance between pLM or gLM embeddings. If the authors assumption holds true that gLM embeddings are more suitable for classifying unannotated proteins, the proposed approach should work at least equally well for the annotated part of proteins.

When comparing gLM embeddings for the paralog matching problem (Fig. 5B, 5C), the authors should add pLM embeddings as reference (most importantly again: quantitative comparison).

One problem that I see with the current function/E.C. analysis is that the authors concatenate pLM and gLM embeddings without benchmarking gLM embeddings in isolation. This can become problematic when training a predictor on top of concatenated embeddings as such a predictor usually has twice as many parameters as the one trained only on pLM embeddings. The additional free parameters increase the risk of overfitting, especially, for redundant datasets as the authors' E.C. dataset which was only reduced at 70% sequence similarity. An alternative to this setup might be to only use embedding-based annotation transfer (similar to homology-based inference but instead of sequence similarity, embeddings-distance is used as means of transferring annotations from a labeled training set to an unlabeled query dataset). Given the relatively small difference between pLM and pLM+gLM embeddings, it would also be important to provide error estimates to allow readers to put the improvement into perspective of statistical significance. (This point, i.e., error estimates, remains true for all reported comparisons).

Additionally, I think it is important to also explain the added complexity of training a new Transformer on top of pLM embeddings. For example, a much simpler approach to compare against would be to simply average over all pLM embeddings in one contig (eg. for 2B and 2E or 5F). I understand that this is not directly comparable because the averaging over all pLM embeddings within one contig is lacking the per-protein resolution but it might give an additional argument to explain why it was necessary to go for a more complex approach. Especially, for Fig. 5F and associated analysis such a comparison might be interesting as the difference in performance might (partly) be attributed to the additional information that is available to gLM. Along the same lines, for analysis presented in Fig. 3D, the authors could also average over pLM embeddings to derive contig embedding. The gLM embeddings should clearly outperform this simple approach meaning that its annotations in the right tail should be less clearly associated with gene transfer.

Minor:

- remove axis ticks from any UMAP plots. Distances in lower-dimensional projection space are not easily interpretable and are rather misleading.
- Increase font size of all text in all plots to improve readability
- Increase dot size for UMAP plots where single point resolution is needed for interpretation (e.g. 2E, 2F, etc)
- Explain choice of clipping ESM embeddings. IF possible, show effect of not clipping. But I understand if this is too compute-intensive.
- Make clear that the LSTM- vs Transformer-comparison is a bit of an apples-against-oranges comparison because the LSTM has orders of magnitude fewer parameters compared to the Transformer.
- Fig. 4C: either add a legend or maybe rather use the 5 CV splits to report confidence/error estimates.

Reviewer #5 (Remarks to the Author):

Reviewer Assessment

Manuscript#: NCOMMS-23-23795-T

Title: Genomic language model predicts protein co-regulation and function

Significance and noteworthy results

The authors have provided a good background on how evolutionary processes shape genomic organization and context in the introduction and have properly motivated the need for models that can learn complex relationships from large genomic datasets. Overall, the manuscript offers an interesting take on linking genomic context and protein sequence-structure-function model through a transformer-based model called the genomic language model (gLM). The problem is significant.

The paper could be strengthened by framing the knowledge gap this work aims to address: What limitations exist with current approaches for analyzing genomic context? How will a language modeling approach for whole metagenomic sequences help overcome these limitations?

A method based on established ESM2

The work aims to build a genomic language model (gLM) based on an existing well-trained pLM (namely ESM2), in other words, to train the transformer-based gLM network to encode genomic information into gLM, using genomic contig data so that it may be used for downstream gene-related tasks. The method may be summarized as follows: A plural of sequences of 15~30 genes in contig is used as the initial training data. The gene sequences are converted into sequences of amino acids and then fed into the ESM2 encoder to yield protein embedding. The embedding is then masked and used as the input to the gLM. The gLM is trained to recover masked embedding. Initial results are provided about the potential use of pLM for integrating protein language models with genomic information.

Experiments and Data Analysis

1. In Figure 3A and Figure 3B, UMAP is utilized to demonstrate the associative power of gLM. However, an excess of noise points may be obscuring the clarity of the visualization. Ideally, the UMAP representation should be consolidated rather than dispersed throughout the two-dimensional space. The authors may preprocess the data to remove outliers before visualizing with UMAP. More importantly, quantitative metrics, such as the KL divergence, should be provided to assess the differences or similarities between representations.

2. Comparative results with the base EMS2: While the paper provides an insightful discussion on contextualized gene embeddings, much of the evidence is presented qualitatively through visualizations. Specifically, experiments may be designed to quantitatively compare the performance of gLM embeddings against the original ESM2 embeddings. A quantitative comparison would provide a more rigorous and convincing assessment of the efficacy and relevance of the gLM embeddings in capturing meaningful genetic contexts.

3. Results on enzyme function prediction: The authors have contrasted the results between pLM and pLM+gLM representations. A setup where only gLM is evaluated might provide insights into whether gLM holds comprehensive information for performing the downstream gene-related tasks effectively on its own, without the aid of pLM. It would further validate the inherent strengths and utility of the gLM embeddings.

4. Regarding Figure 3(D): The current results show the outcomes based on contextualized embeddings. For a comprehensive understanding and fair comparison, results derived from context-free embeddings, in addition to those from the contextualized ones, would provide valuable information. This way can help readers assess the relative advantages or differences between the two methods more clearly.

Soundness and Potential Impact

The paper claims to have demonstrated that the gLM model captures meaningful biological signals related to gene function, regulation, and taxonomy. However, providing results on how the gLM model has improved over the base ESM-2 quantitatively can be more direct and valuable. For example, in Figure 2, it would be informative to provide visualization of the context-free protein embeddings from ESM-2 to illustrate better what new insights are gained from gLM contextualization. In Figure 3C, comparing gLM to ESM-2 in EC number prediction would help benchmark performance gains.

Questions also arise about how much genomic knowledge has been embedded and whether the gLM can be called a gene model. The gLM model does not directly encode gene-related knowledge except using the gene-related contig data at the front end.

Therefore, the impact on the related field is limited due to insufficient experimental evidence.

Minor questions

1. While the number of genes in a contig can exceed 30, why was 30 chosen as the maximum value? Any experimental results to support the choice?
2. In Figure 1b, the 'gLM' module is better described as the 'gLM encoder,' and the output may be a reconstructed representation.
3. In Figure 5, parts (D) and (E): What UMAP parameters are chosen specifically for the two visualizations? It is better to provide the numbers for better interpretation.

Reproducibility

The code is provided. It should be easy to reproduce.

Reviewer #1 (Remarks to the Author):

Author responses

Hwang's paper "Genomic language model predicts protein co-regulation and function" develops a genomic language model (gML) to predict protein function based on operon structure. The model is impressive and well thought out. I only have a few minor comments that need to be addressed:

We thank the reviewer for the kind feedback.

1) The operon framework that the gLM relies on only works with prokaryotes and their viruses. The authors need to openly address this.

Thank you for suggesting this. We have included a sentence in the corresponding results section in L284 "Operons are prevalent in bacterial, archaeal and their viral genomes, while rare in eukaryotic genomes.". We also included a sentence in L450 in the discussion section: "Finally, our model was trained largely on bacterial, archaeal and viral genomes, therefore, how this method can be adapted for eukaryotic genomes with added complexity remains to be further explored."

2) The authors miss key contributors to the context-based ML field including PMID: 31520072.

Thank you, we have added this citation along with other works in the field in the introduction in L40.

Reviewer #2 (Remarks to the Author):

Hwang et al. present a genomic language model that encodes genes via embeddings of a protein language model that get contextualised by neighbouring genes on metagenomic contigs of 15-30 genes.

The hypothesis of the manuscript is valuable and clear: protein embeddings decontextualised of their genomic environment will miss genomic context important for predictions in gene world. Protein language models encode a demonstrated notion of protein structure and function which can sufficiently represent genes. The authors execute well on the analysis of decontextualised vs. contextualised gene/protein representations with several interesting examples.

We thank the reviewer for the kind feedback.

Main observations:

- In my opinion, calling the model a "genomic language model" is potentially misleading. Protein language models operate on protein sequences treated as "language" through their building blocks, amino acids. A genomic language model, in my opinion, is a model that operates on nucleotides. The model presented here doesn't explicitly account for nucleotides, making it rather a gene order aware protein language model than a genomic language model, as is also argued in 43-55 of introduction. In my opinion this is important because, for instance, synonymous mutations will be transparent to the model.

We refer to this model as a "genomic language model" because we model the co-evolutionary relationships between genes in a genome, where such relationships are generalizable across organisms. We believe that the field's definition of "genomics" focuses on the inter-relationships between genes and their resulting function. For instance, the use of the term "genome" to describe the collection of genes predates the discovery of nucleotides as the basis of genetic

material(Cristescu 2019). Additionally, per WHO's definition: “genomics addresses all genes and their inter relationships in order to identify their combined influence on the growth and development of the organism.” Therefore, we call this model that operates on gene-level representations a genomic language model. The suggested name “gene-order aware protein language model” does not capture the essence of our model because 1) it is not a protein language model, as we operate on multiple genes (one token = one gene) 2) it is not simply order-aware because the model attends to rich representations of other protein-coding genes and is designed to learn the functional interactions between genes in a given contig.

However, we acknowledge that by using protein representations, we abstract away the information available at the nucleotide level. This design choice is based on the assumption that most synonymous mutations result in no change in the resulting protein's primary function. We are aware that recent studies have shown the effects of some synonymous mutations on the protein's expression level. As expression levels of proteins is beyond the scope of this study, we decided to exclude the effects of synonymous mutations, in order to gain greater context length and leverage rich pLM representations. We acknowledge that future research should account for synonymous mutations as we expand the scope of gLM and we note this future direction in L442: “Future iterations of the model could use raw residue-level or codon-level embeddings as input to allow modeling of residue-to-residue co-evolutionary interactions between proteins and synonymous mutation effects on gene function.”

Furthermore, we acknowledge that our model does not operate on non-coding DNA (ncDNA) between genes. This is a limitation of our study and we elaborate further on this in our discussions as a clear future research direction in L448. We posit that our model nevertheless captures a large majority of “genomic” interactions in microbial genomes, where more than 90% of the DNA is coding. Admittedly, this is not true for eukaryotic genomes that feature much larger fractions of intergenic elements. We note this limitation in the manuscript in L450: “Finally, our model was trained largely on bacterial, archaeal and viral genomes, therefore, how this method can be adapted for eukaryotic genomes, especially those with extensive intergenic regions, remains to be further explored.”

- I wonder how this approach stands compared to the likes of Enformer (Avsec et al., 2021), DNABert (Ji et al. 2021), or more recent work like GenSLMs (Zvyagin, 2022), CaLM (Outeiral, 2022) and Nucleotide Transformer (Dalla-Torre et al., 2023), all encoding some notion of "genomic language model", some explicitly trying to address the (long-)context issue (Enformer, GenSLMs & Nucleotide Transformer). Especially interesting in this context is GenSLMs, which includes similar principles to this work: a foundational language modelling piece, yet operating on codons, and a contextualising piece, yet based on diffusion. [Note: conflicting interest on Dalla-Torre and Zvyagin]

We thank the reviewer for mentioning these previous works. We have now added an introductory section that compares gLM to these recent efforts in L47-57:

“On the other end of the spectrum of representations, there have been efforts to use unsupervised learning on nucleotide sequences to predict gene expression level¹⁸ and detect regulatory motifs¹⁹⁻²¹. These models are largely trained and benchmarked on the human genome and focus on predicting gene regulation rather than function. Most recent efforts to leverage diverse microbial sequences to model genome-scale information include GenSLMs²², which is pretrained on codon-level representations of diverse bacterial and viral gene sequences and later fine-tuned on SARS-CoV-2 genomes. In order to learn generalizable gene-to-gene-context interactions across biology, a model needs to be pretrained on 1) diverse lineages of organisms, 2) rich and continuous representation of genes and 3) longer segments of genomes with multiple genes. To

our knowledge, there has been no method that combines all three aspects of pretraining to learn genomic information across diverse lineages of biology (see summary of previous studies in **Extended Data 1**).”

While the above mentioned models are similar to gLM in that they operate on fragments of genomes, gLM differs in that 1) it is pretrained on genomic data across many organisms, and 2) its pretraining task learns interactions between multiple genes. This allows gLM to learn generalizable and organism-agnostic co-evolutionary interactions across genes.

- Unlike gLM, Enformer and DNABert are trained only on the human genome, and nucleotide transformer is trained with a significant bias on human genomes. These models were trained to learn gene expression and regulation dynamics in eukaryotic genomes (particularly in the human genome), while gLM is trained to learn co-evolutionary interactions between genes in genomic sequences.
- GenSLM pretraining masks codons within a single gene (similar to pLMs) and therefore cannot learn interactions between genes during pretraining. The GenSLM’s genome-scale model is fine-tuned only on SARS-CoV-2 genomes, and therefore is not generalizable/organism-agnostic.
- CaLM is a protein language model, so we cite this work alongside other pLMs in L33.

We also paste below the newly added Extended Data 1 that compares gLM to previous approaches.

Extended Data 1. Comparison of gLM to previous efforts in modeling various aspects of biological sequences.

	Multi-gene input	Continuous representation of genes	Generalizable across organisms (Organism-agnostic pretraining)	Self-supervised language model
gLM (this study)	✓	✓	✓ (Metagenomic sequences with bias towards bacteria, archaea and viruses)	✓
pLMs ⁴⁻⁷ (e.g. ESM2, ProtBert, ProtT5, ProGen, CaLM)	✗	✓	✓	✓
Miller et al ¹⁶	✓	✗	✓	✗
Enformer ¹⁸	✓	✓	✗ (Pretrained on human and mouse genomes only)	✗
DNABERT ¹⁹	✗ (Max context length of DNABERT-6 is 3072 bp, which is not sufficient to include a median length (26,288 bp) human protein coding gene ⁷⁰)	✓	✗ (Pretrained on human genome)	✓
Nucleotide Transformer ²⁰	✗ (Max context length is 6000 bp, which is not sufficient to include a median length (26,288 bp) human protein coding gene)	✓	✗ (Heavily biased towards human genome)	✓

HyenaDNA ²¹	✓	✓	✗ (Pretrained on Human genome)	✓
GenSLM-foundation model ²²	✗ (Single genes used for pretraining)	✓	✓	✓
GenSLM-SARS-CoV2 genome model ²²	✓	✓	✗ (fine-tuned on SARS-CoV2 genomes only)	✓

- Not a lot of explanation or ablation was given to the choice of PCA as reduction technique for ESM2 representations, and for the "parameter" choices, e.g. PCA up to component 100 (why not 10? Why not 500?). Does 100 include 90% of the variation? What would happen if only 70 or 40% of variation is considered, how does it affect downstream performance?

Thank you for pointing this out. We decided to use PCA on labels because the reduction of label dimensionality helped with the stability of training. We chose to use 99 PCs (+ orientation feature = 100 features) for pLM representations because 89.7% of the variances were captured in the first 99 PCs and we saw a drop in the cumulative explained variances explained after ~100 PCs out of 1280 PCs as shown below:

Extended Data 15. Cumulative explained variance of principal components of ESM2 embeddings calculated on 2.5 million randomly selected MGYPs.

We have now added this as Extended Data 15 and we have added the total explained variance in the methods section in L498, as well as the reasoning for using PCA.

- I think an important baseline would be a model encoding genes with tokenises, other categorical representations, or simple features like length, nucleotide or codon counts (bag-of-words), etc. At current, it's impossible to asses whether the signal from pLMs influences the downstream predictions.

We thank the reviewer for this suggestion. The use of continuous and rich representations from pLM as input is an important component of training gLM, which is now demonstrated by ablating pLM representations. We conducted an ablation of gLM with mean-pooled one-hot representations of the protein sequences instead of mean-pooled pLM representations. We trained until the model converged after 57 epochs (<0.1% decrease in loss over 40k iterations). We compare the performance of the model through validation accuracy, where it performs equivalent to random. We also compare the attention-based operon prediction accuracies, where it performs significantly worse (~45% reduction in mean average precision). We report the results of this ablation in Extended Data 5 and refer to this in L101 and L294.

Extended Data 5. Ablation of pLM representations. Ablated gLM was trained on one-hot representations until convergence (<0.1% decrease in loss over 40k iterations).

	gLM	gLM one-hot
Representations	ESM2 embedding	One-hot amino acid encoding
Pooling	mean	mean
Number of layers	19	19
Attention heads	10	10
Input embedding dimension	1281	34
Hidden size	1280	1280
Batch size	3000	3000
Learning rate	1e-4	1e-4
Warm up steps	5000	5000
Training steps	1296960	132050
Number of predictions	4	4
Number of parameters	954736916	945338764
% Pseudo-accuracy (validation)	71.9	3.29
% Absolute accuracy (validation)	59.2	0.002
Operon prediction mAP	0.775 ± 0.29	0.426 ± 0.015

While pLM representation of genes is critical for training gLM, gLM learns important contextual information using co-evolutionary signals across multiple protein sequences. Therefore, as to the reviewer's second point regarding pLM signals, we fully expect pLM signals to propagate through gLM and result in meaningful contextualization of the gene. In our analyses (e.g. Figure 2EF, 2HI, 4EF, 5DE) we compare input protein embeddings (ESM2 representations) to contextualized

protein embeddings (gLM representations) to assess how much information gLM can learn from contextualizing pLM embeddings. We also conducted an additional EC number classification on masked protein embeddings (Extended Data 8, where the pLM input for the queried protein is masked, in order to explicitly quantify how much functional information can be captured purely from the context alone.) We show that even when pLM information of the queried protein is unavailable to gLM, gLM can rely on the protein's context alone to generate meaningful representations that have functional significance for the queried protein. We elaborate more on this in L220-241 and below in our response to the reviewer's question on the EC prediction task.

- Similarly, the authors didn't include an ablation of nucleotide transformer models (mentioned earlier), instead of protein representations. Arguably, this models will have higher sensitivity towards nucleotide changes, which may be the goal of a genomic language model.

We thank the reviewer for this suggestion. The goal of our genomic language model is to learn the primary functions of genes (e.g. enzyme class) and their functional modules (e.g. operons) encoded in genomes. Therefore, detection of fitness effects (e.g. expression levels) resulting from synonymous mutations falls beyond the scope of our study.

Furthermore, while an interesting ablation, there currently exists no appropriate nucleotide transformer model that can be used to represent genomic sequences in our corpus. Previous transformer models on nucleotide sequences (e.g. Enformer, DNABert, Nucleotide Transformer) are trained exclusively or almost entirely on human genome sequences, where only ~1% of the genome consists of coding sequences. Microbial genomes, on the other hand, are much denser in coding sequences (~85-90%) (PMID: 22064560). Therefore, representing microbial genes using an encoder largely trained on human intergenic regions will lead to poor representations and train-inference mismatch.

Considering these reasons, along with the computational cost of training a new model from scratch (~6-8 weeks), we hope the reviewer understands our decision to forego this ablation.

- Fig 3 A: I would strongly discourage the use of "cluster distinctly" for these plots, as it's visually impossible to verify, practically questionable (maybe a different UMAP parametrisation would lead to opposite results), and statistically not substantiated.

We have removed these UMAP figures, and instead replaced them with quantitative measures of KL divergence between unannotated and annotated fractions of data. The new analysis is added to L206:

"We compared the distributions of unannotated and annotated fractions of proteins in our dataset using context-free pLM embeddings and contextualized gLM embeddings. We found a statistically significant lower divergence between distributions of unannotated and annotated genes in gLM embeddings compared to pLM embeddings (paired t-test of Kullback-Leibler divergences, t-test statistic = 7.61, p-value < 1e-4, n = 10; see Methods for sampling and metric calculation)."

The method section (L540) corresponding to this calculation is pasted below:

"Distributions of unannotated and annotated embeddings

Distributions of unannotated and annotated embeddings in the database were compared using Kullback-Leibler (KL) divergence analysis. First, ten random samples of 10,000 subcontigs from the MGnify corpus. pLM and gLM embeddings of the genes were calculated using mean-pooled last hidden layer of ESM2 embeddings and mean-pooled last hidden layer of gLM respectively. Outliers were removed using Mahalanobis distance and a chi-squared threshold of 0.975. pLM and gLM embedding dimensions were reduced to 256 principal components (91.9 ± 1.72% and 80.1 ± 6.89% total variances explained respectively). KL divergence was calculated using the following equation:

$$D_{KL}(P||Q) = \frac{1}{2}(tr(\Sigma_1^{-1}\Sigma_0) - k + (\mu_1 - \mu_0)^T \Sigma_1^{-1}(\mu_1 - \mu_0) + \ln(\frac{\det \Sigma_1}{\det \Sigma_0}))$$

, where P corresponds to the distribution of unannotated genes and Q corresponds to the distribution of annotated genes, with μ_0, μ_1 respectively as means and Σ_0, Σ_1 respectively as covariance matrices. The significance of the KL divergence differences between pLM and gLM embeddings is calculated using a paired t-test across the ten samples.”

- Line 202 the authors argue about a 2-3% increase in prediction accuracy. Is this within std. error? Regarding the dataset, my understanding from 513 is that the data is split randomly? Minimally, considering the traditional protein sequence context, train/test should be split according to sequence redundancy. Given that in this case the objective is the generalisation of genomic context, other and/or additional considerations should be included to make sure that the downstream model isn't leveraging leakage of information, e.g. similar genomic arrangement or content between train and test contigs. (Train sample: Gene 1, Gene 2, Gene 3 -> Gene 2 -> EC; Test sample: Gene 3, Gene 2, Gene1 -> Gene 2 -> EC).

*Thank you for the suggestion. We redid the EC number prediction analysis to incorporate the suggestions by all reviewers. 1) We reduced the sequence identity threshold for train-test split to 30% sequence identity to minimize leakage. Because no gene in the test set is >30% similar to genes in the train set, the provided scenario (Train sample: Gene 1, Gene 2, Gene 3 -> Gene 2 -> EC; Test sample: Gene 3, Gene 2, Gene1 -> Gene 2 -> EC) will not occur. 2) We calculated the standard error and compared statistically significant differences in expressiveness of pLM and gLM embeddings. 3) We compare predictors trained on pLM and gLM embeddings separately (instead of concatenated embeddings). Notably, in **Extended Data 8**, we compared pLM (context-free) embedding, gLM embedding where the queried protein is masked at the time of inference and therefore only context is utilized (context-only gLM embedding) and gLM embedding without masking (contextualized gLM embedding). In doing so, we show that masked gLM embedding, even without any pLM information, carries important information that can be leveraged for enzyme function prediction.*

Please find the updated results section and figures are below:

Contextualization improves enzyme function prediction

To test the hypothesis that genomic context of proteins can be used to aid function prediction, we compared how much the addition of context information can improve the expressiveness of protein representations for enzyme function prediction. First, we generated a custom MGY-EC dataset where the train and test data were split at 30% sequence identity for each EC class (see **Methods**). Second, we apply a linear probe (LP) to compare the expressiveness of representations at each gLM layer, with and without masking the queried protein (**Extended Data 8**). By masking the queried protein, we can assess gLM's ability to learn functional information of a given protein, only from its genomic context, without the propagation of information from the protein's pLM embeddings. We observed that a large fraction of contextual information pertaining to enzymatic function is learned in the first six layers of gLM. We also demonstrate that context information alone can be predictive of protein function, reaching up to $24.4 \pm 0.8\%$ accuracy. In contrast, without masking, gLM can incorporate information present in the context with the original pLM information for each queried protein. We observed an increase in expressivity of gLM embeddings also in the shallower layers, with accuracy reaching up to $51.6 \pm 0.5\%$ in the first hidden layer. This marks a $4.6 \pm 0.5\%$ increase from context-free pLM prediction accuracy (**Figure 3A**) and mean average precision (**Figure 3C**) Thus, we demonstrate that information that gLM

learns from the context is orthogonal to information captured in pLM embedding. We also observed diminishing expressivity in enzyme function information with deeper layers of gLM; this reflects the masked pretraining objective that is independent of enzyme function prediction task and is consistent with previous examinations of LLMs, where specific layers perform better than others for downstream tasks. Finally, to further examine the expressiveness of these representations, we compared per-class F1 score gains (**Figure 3B**). We observe statistically significant differences in F1 scores (*t*-test, Benjamini/Hochberg corrected *p*-value < 0.05) between the two models in 36 out of 67 EC classes with more than ten samples in the test set. Majority (27 out of 36) of the statistical differences resulted in improved F1 score in MLP trained on gLM representations.

Figure 3. Contextualization of unannotated proteins. **A)** Linear probe EC classification accuracy for pLM(ESM2) representations and gLM (1st hidden layer) representations. **B)** F1-score comparisons of statistically significant (Benjamini/Hochberg corrected *p*-value < 0.05) differences in performance of pLM- and gLM-based EC number linear probes. EC classes are ordered with the largest gain with contextualization on the left to the largest loss with contextualization on the right. **C)** Precision-Recall curves of pLM- and gLM-based EC number linear probes. **D)** Histogram of variance (# bins = 100) calculated using contextualized embeddings (gLM; orange) and contig-averaged pLM (blue) embeddings of MGYPs that occur at least 100 times in the database. Histograms for unannotated and annotated fraction of the MGYPs are plotted separately and bars are not stacked. Annotated examples in the long right tail include

phage proteins and transposases, reflecting their ability to self-mobilize (see annotations of top tens most variant genes in **Extended Data 9**).

A

B

C

D

Extended Data 8. Linear probing of context-free, context-only and contextualized gene embeddings. A) Schematics of how context-free, context-only, and contextualized gene embeddings are extracted for linear probing. Context-only gene embeddings are extracted by

masking the queried gene, therefore the original pLM signal cannot propagate through to gLM representation. This can be used to quantify how gLM-learned context informs EC number prediction independent of pLM signal. B) Schematics of how LP is used for EC number classification. C) Per-layer linear probing accuracy of gLM contextualized embeddings, where no-masking was performed at the time of inference. The 0th layer of gLM is equivalent to context-free pLM embedding. D) Per-layer linear probing accuracies of gLM context-only embeddings, where the queried genes are masked at the time of inference.

Other:

- Reference 50 may need amendment

We have now amended this reference. Thank you.

- 447-448 "ESM2 embeddings were 448 subsequently standardised and clipped at (-10,10)" -- What's "standardised" or "clipped"?

We have now edited this sentence to L483 "ESM2 embeddings were normalized (by subtracting the mean of each feature and dividing by its standard deviation) and clipped such that all features range from -10 to 10."

[Christian Dallago]

Reviewer #3 (Remarks to the Author):

The authors present a new language model that bridges the gap between genomic data and proteins. For this, they use embeddings derived from a pre-trained protein language model (pLM), ESM-2, as input to a newly trained language model (encoder-architecture; (Ro)BERT(a)-like), dubbed gLM (genomic language model). Instead of training either on nucleic or amino acids, the authors mix both modalities by deriving exon-structure from millions of metagenomic scaffolds, represented by per-protein ESM-2 embeddings (average-pooling applied over the length dimension of the protein) which are used to train the proposed genomic language model (gLM) on reconstructing corrupted genes using non-corrupted (genomic) context. Simply put, the authors put per-protein embeddings into their genomic context. After training, the authors benchmark their gLM on a variety of tasks, i.e., enzymatic function, taxonomy, and operon structure. The proposed method has the potential to set the foundation for many novel applications. For example, one can derive embeddings that allow to distinguish proteins with identical sequences based on their genomic context. The paper is well-written with many figures illustrating the major arguments of the authors.

We thank the reviewer for the kind feedback.

The major problem that I see with the current form of the paper is that a large fraction of its evaluation is based on qualitative analysis of UMAP-projections. Despite being unquestionable interesting, low-dimensional projections s.a. t-SNE or UMAP are susceptible to parameter choice and one should be careful when interpreting the global structures in this space. For example, the analysis of 2E and 2F, or 2H and 2I, or 4E and 4F and 5D and 5F relies mostly on the global structure output by UMAP which can change severely when changing parameters. A more quantitative way could be to compute clustering metrics s.a. silhouette scores for those cases using e.g. Euclidean/Cosine distance in gLM/pLM embedding space. This allows for a quantitative comparison of the methods and should also allow to quantify which signal is captured by the orthogonal signals captured by pLMs and gLMs, i.e., silhouette score computed for data in 2H

and 2l once based on structural aspects and once on functional aspects (phage defense proteins cluster etc).

Thank you for this suggestion. We have made following changes to address this feedback:

- *We have now added silhouette scores to all UMAP-projections in Figure 2 and refer to them in the main text.*
- *We have removed UMAP-projections for the previous Figure 3AB, and replaced it with KL divergence calculation as suggested by reviewer 5. Please see our answer below that discusses this analysis further.*
- *We provide KL divergence metric as well as hierarchical clustering dendrogram for Figures 4EF. We chose to calculate KL divergence and visualize hierarchical clustering of the embeddings for Figure 4EF, because we have sufficient data points for KL divergence calculation, and we are interested in the hierarchical structure of clustering which would not be easy to capture through silhouette score analysis.*
- *We provide a logistic regression classification metric for Figure 5DE. We chose to use logistic regression classifications for Figure 5 because we have sufficient data points for these figures, and logistic regression directly captures the expressiveness of these learned high dimensional representations and provides information on the capabilities for task-specific transfer learning or fine-tuning.*

The same, albeit more complex because of a lack of labels, holds true for e.g. 3A and 3B. Despite the lack of labels, the authors could still compute metrics on transferring annotations from labeled/predicted lookup sets to labeled/predicted query proteins using distance between pLM or gLM embeddings. If the authors assumption holds true that gLM embeddings are more suitable for classifying unannotated proteins, the proposed approach should work at least equally well for the annotated part of proteins.

We have removed these UMAP figures, and instead replaced them with quantitative measures of KL divergence between unannotated and annotated fractions of data, as suggested by reviewer 5. The new analysis is added to L206:

“We compared the distributions of unannotated and annotated fractions of proteins in our dataset using context-free pLM embeddings and contextualized gLM embeddings. We found a statistically significant lower divergence between distributions of unannotated and annotated genes in gLM embeddings compared to pLM embeddings (paired t-test of Kullback-Leibler divergences, t-test statistic = 7.61, p-value < 1e-4, n = 10; see Methods for sampling and metric calculation).”

The method section corresponding to this calculation is pasted below:

“Distributions of unannotated and annotated embeddings

Distributions of unannotated and annotated embeddings in the database were compared using Kullback-Leibler (KL) divergence analysis. First, ten random samples of 10,000 subcontigs from the MGNify corpus. pLM and gLM embeddings of the genes were calculated using mean-pooled last hidden layer of ESM2 embeddings and mean-pooled last hidden layer of gLM respectively. Outliers were removed using Mahalanobis distance and a chi-squared threshold of 0.975. pLM and gLM embedding dimensions were reduced to 256 principal components (91.9 ± 1.72% and 80.1 ± 6.89% total variances explained respectively). KL divergence was calculated using the following equation:

$$D_{KL}(P||Q) = \frac{1}{2}(\text{tr}(\Sigma_1^{-1}\Sigma_0) - k + (\mu_1 - \mu_0)^T \Sigma_1^{-1}(\mu_1 - \mu_0) + \ln(\frac{\det \Sigma_1}{\det \Sigma_0}))$$

, where P corresponds to the distribution of unannotated genes and Q corresponds to the distribution of annotated genes, with μ_0, μ_1 respectively as means and Σ_0, Σ_1 respectively as covariance matrices. The significance of the KL divergence differences between pLM and gLM embeddings is calculated using a paired t-test across the ten samples.”

When comparing gLM embeddings for the paralog matching problem (Fig. 5B, 5C), the authors should add pLM embeddings as reference (most importantly again: quantitative comparison).

Paralog matching is conducted using zero-shot gLM predictions. The pre-training task of gLM is to predict the masked gene given its context. We tested the ability of gLM to predict the embedding of the correctly paired paralog, by masking one of the genes in a pair of paralogs. Since this prediction task is directly derived from the gLM pretraining task, we cannot baseline this using pLM embeddings (pLM's pretraining was done on masked amino acid residue prediction). Therefore, we quantify the significance gLM's performance against simulated random data. Please find the relevant methods section pasted below:

"We constructed test contigs consisting of three genes, where first and third genes are masked, and the second gene encodes one of the pair in forward direction. We then queried gLM to predict the two neighboring masked genes, and considered the prediction to be correct if either of the proteins closest to masked genes's highest confidence prediction in embedding space belongs to the same sequence cluster as the interacting protein (50% amino acid sequence identity, calculated using CD-HIT v4.6⁶⁹). Random chance correct prediction rate was simulated using 1000 iterations of random predictions generated within the standard normal distribution and performing the same operation as above to compute the rate of correct predictions."

One problem that I see with the current function/E.C. analysis is that the authors concatenate pLM and gLM embeddings without benchmarking gLM embeddings in isolation. This can become problematic when training a predictor on top of concatenated embeddings as such a predictor usually has twice as many parameters as the one trained only on pLM embeddings. The additional free parameters increase the risk of overfitting, especially, for redundant datasets as the authors' E.C. dataset which was only reduced at 70% sequence similarity. An alternative to this setup might be to only use embedding-based annotation transfer (similar to homology-based inference but instead of sequence similarity, embeddings-distance is used as means of transferring annotations from a labeled training set to an unlabeled query dataset). Given the relatively small difference between pLM and pLM+gLM embeddings, it would also be important to provide error estimates to allow readers to put the improvement into perspective of statistical significance. (This point, i.e., error estimates, remains true for all reported comparisons).

*Thank you for the suggestion. We redid the EC number prediction analysis to incorporate the suggestions by all reviewers. The major changes are as follows: 1) We reduced the sequence identity threshold for train-test split to 30% to minimize leakage. 2) We compare predictor trained on pLM and gLM embeddings separately (instead of concatenated embeddings). Notably, in **Extended Data 8**, we compared pLM (context-free) embedding, gLM embedding where the queried protein is masked at the time of inference and therefore only context is utilized (context-only gLM embedding) and gLM embedding without masking (contextualized gLM embedding). In doing so, we show that masked gLM embedding, even without any pLM information, carries important information that can be leveraged for enzyme function prediction. 3) We conducted linear probing on different layers in gLM separately for EC number prediction and show that embeddings derived from layer 1 carries the most EC relevant information. 4) We added error estimates for linear probing results. Please find the updated methods section in L554-573:*

And the updated results section and figures are below:

Contextualization improves enzyme function prediction

To test the hypothesis that genomic context of proteins can be used to aid function prediction, we compared how much the addition of context information can improve the expressiveness of protein representations for enzyme function prediction. First, we generated a custom MGYE-EC dataset where the train and test data were split at 30% sequence identity for each EC class (see

Methods). Second, we apply a linear probe (LP) to compare the expressiveness of representations at each gLM layer, with and without masking the queried protein (**Extended Data 8**). By masking the queried protein, we can assess gLM's ability to learn functional information of a given protein, only from its genomic context, without the propagation of information from the protein's pLM embeddings. We observed that a large fraction of contextual information pertaining to enzymatic function is learned in the first six layers of gLM. We also demonstrate that context information alone can be predictive of protein function, reaching up to $24.4 \pm 0.8\%$ accuracy. In contrast, without masking, gLM can incorporate information present in the context with the original pLM information for each queried protein. We observed an increase in expressivity of gLM embeddings also in the shallower layers, with accuracy reaching up to $51.6 \pm 0.5\%$ in the first hidden layer. This marks a $4.6 \pm 0.5\%$ increase from context-free pLM prediction accuracy (**Figure 3A**) and $5.5 \pm 1.0\%$ increase in mean average precision (**Figure 3C**). Thus, we demonstrate that information that gLM learns from the context is orthogonal to information captured in pLM embedding. We also observed diminishing expressivity in enzyme function information with deeper layers of gLM; this is consistent with previous examinations of LLMs, where deeper layers are specialized to the pretraining task (masked token prediction), and is consistent with previous examinations of LLMs, where the best-performing layer depends on the specific downstream tasks (Rogers et al.). Finally, to further examine the expressiveness of these representations, we compared per-class F1 score gains (**Figure 3B**). We observe statistically significant differences in F1 scores (t-test, Benjamini/Hochberg corrected p-value < 0.05) between the two models in 36 out of 67 EC classes with more than ten samples in the test set. Majority (27 out of 36) of the statistical differences resulted in improved F1 score in MLP trained on gLM representations.

Figure 3. Contextualization of unannotated proteins. **A)** Linear probe EC classification accuracy for pLM(ESM2) representations and gLM (1st hidden layer) representations. **B)** F1-score comparisons of statistically significant (Benjamini/Hochberg corrected p -value < 0.05) differences in performance of pLM- and gLM-based EC number linear probes. EC classes are ordered with the largest gain with contextualization on the left to the largest loss with contextualization on the right. **C)** Precision-Recall curves of pLM- and gLM-based EC number linear probes. **D)** Histogram of variance (# bins = 100) calculated using contextualized embeddings (gLM; orange) and contig-averaged pLM (blue) embeddings of MGYPs that occur at least 100 times in the database. Histograms for unannotated and annotated fraction of the MGYPs are plotted separately and bars are not stacked. Annotated examples in the long right tail include phage proteins and transposases, reflecting their ability to self-mobilize (see annotations of top tens most variant genes in **Extended Data 9**).

A

B

D

Extended Data 8. Linear probing of context-free, context-only and contextualized gene embeddings. A) Schematics of how context-free, context-only, and contextualized gene embeddings are extracted for linear probing. Context-only gene embeddings are extracted by masking the queried gene, therefore the original pLM signal cannot propagate through to gLM representation. This can be used to quantify how gLM-learned context informs EC number

prediction independent of pLM signal. B) Schematics of how LP is used for EC number classification. C) Per-layer linear probing accuracy of gLM contextualized embeddings, where no-masking was performed at the time of inference. The 0th layer of gLM is equivalent to context-free pLM embedding. D) Per-layer linear probing accuracies of gLM context-only embeddings, where the queried genes are masked at the time of inference.

Additionally, I think it is important to also explain the added complexity of training a new Transformer on top of pLM embeddings. For example, a much simpler approach to compare against would be to simply average over all pLM embeddings in one contig (eg. for 2B and 2E or 5F). I understand that this is not directly comparable because the averaging over all pLM embeddings within one contig is lacking the per-protein resolution but it might give an additional argument to explain why it was necessary to go for a more complex approach. Especially, for Fig. 5F and associated analysis such a comparison might be interesting as the difference in performance might (partly) be attributed to the additional information that is available to gLM.

*Thank you for the suggestion. We have visualized and calculated silhouette scores for contig-averaged pLM embeddings for 2B,E,H in **Extended Data 7**. We added the following section (L166-172) to refer to these additional results: “In order to quantify the information gain as a result of training a transformer on genomic contexts, we compare clustering results in 2B, F, and I with clustering conducted on (sub)contig-averaged pLM embeddings (**Extended Data 7**). By mean-pooling pLM embeddings across a given subcontig, we can summarize the context information as a naive baseline. We report more consistent clusterings (higher silhouette scores) of gLM embeddings compared to contig-averaged pLM for all three analyses (see **Extended Data 7** figure captions for values). We demonstrate that the gLM transformer model learns representations that correlate with biological function, which are not captured by the naive baseline.”*

*Additionally, we conducted the same analysis for AAA+ regulator analysis (**Figure 4DEF** and **Extended Data 11B**) and taxonomic analysis (**Figure 5DEF**), where we show an improved clustering and higher silhouette scores with gLM embeddings compared to contig-averaged pLM embeddings.*

Along the same lines, for analysis presented in Fig. 3D, the authors could also average over pLM embeddings to derive contig embedding. The gLM embeddings should clearly outperform this simple approach meaning that its annotations in the right tail should be less clearly associated with gene transfer.

*Thank you for the suggestion. We have now updated Figure 3D to compare variances in contig averages of pLM embeddings with variances in gLM embeddings and report the annotations of the top most variant genes found using the two metrics in **Extended Data 9**.*

Minor:

- remove axis ticks from any UMAP plots. Distances in lower-dimensional projection space are not easily interpretable and are rather misleading.

We have removed axis ticks from all UMAP plots.

- Increase font size of all text in all plots to improve readability

We have increased the font size in all plots.

- Increase dot size for UMAP plots where single point resolution is needed for interpretation (e.g. 2E, 2F, etc)

We have increased dot sizes.

- Explain choice of clipping ESM embeddings. IF possible, show effect of not clipping. But I understand if this is too compute-intensive.

We chose to clip ESM embeddings because we noticed that some ESM2 embedding features were very large in magnitude even after standardization. Without clipping, we observed that training was unstable. We now state this in the methods section in L485.

- Make clear that the LSTM- vs Transformer-comparison is a bit of an apples-against-oranges comparison because the LSTM has orders of magnitude fewer parameters compared to the Transformer.

Thank you for pointing this out. We note this now in the manuscript and also added the explanation of why a smaller biLSTM was used in L99 and in the caption of Extended Data 4: "note that biLSTM is smaller because it failed to converge when scaling the number of layers".

- Fig. 4C: either add a legend or maybe rather use the 5 CV splits to report confidence/error estimates.

We have now added a legend for each split.

Reviewer #5 (Remarks to the Author):

Reviewer Assessment

Manuscript#: NCOMMS-23-23795-T

Title: Genomic language model predicts protein co-regulation and function

Significance and noteworthy results

The authors have provided a good background on how evolutionary processes shape genomic organization and context in the introduction and have properly motivated the need for models that can learn complex relationships from large genomic datasets. Overall, the manuscript offers an interesting take on linking genomic context and protein sequence-structure-function model through a transformer-based model called the genomic language model (gLM). The problem is significant.

The paper could be strengthened by framing the knowledge gap this work aims to address: What limitations exist with current approaches for analyzing genomic context? How will a language modeling approach for whole metagenomic sequences help overcome these limitations?

*We thank the reviewer for these suggestions. We have now added an introductory section that compares gLM to these existing efforts in L43-57 and how the use of metagenomic sequences consisting of multi-gene segments from diverse lineages of biology addresses the current knowledge gap. We also summarize the this knowledge gap as a table in **Extended Data 1**:*

*“Recent efforts to model genomic information have shown predictive power of genomic context in gene function¹⁶ and metabolic trait evolution¹⁷ in bacterial and archaeal genomes. However, these methods represent genes as categorical entities, despite these genes existing in continuous space where multidimensional properties such as phylogeny, structure, and function are abstracted in their sequences. On the other end of the spectrum of representations, there have been efforts to use unsupervised learning on nucleotide sequences to predict gene expression level¹⁸ and detect regulatory motifs¹⁹⁻²¹. These models are largely trained and benchmarked on the human genome and focus on predicting gene regulation rather than function. Most recent efforts to leverage diverse microbial sequences to model genome-scale information include GenSLMs²², which is pretrained on codon-level representations of diverse bacterial and viral gene sequences and later fine-tuned on SARS-CoV-2 genomes. In order to learn generalizable gene-to-gene-context interactions across biology, a model needs to be pretrained on 1) diverse lineages of organisms, 2) rich and continuous representation of genes and 3) longer segments of genomes with multiple genes. To our knowledge, there has been no method that combines all three aspects of pretraining to learn genomic information across diverse lineages of biology (see summary of previous studies in **Extended Data 1**).“*

Extended Data 1. Comparison of gLM to previous efforts in modeling various aspects of biological sequences.

	Multi-gene input	Continuous representation of genes	Generalizable across organisms (Organism-agnostic pretraining)	Self-supervised language model
gLM (this study)	✓	✓	✓ (Metagenomic sequences with bias towards bacteria, archaea and viruses)	✓
pLMs ⁴⁻⁷ (e.g. ESM2, ProtBert, ProtT5, ProGen, CaLM)	✗	✓	✓	✓

Miller et al ¹⁶	✓	✗	✓	✗
Enformer ¹⁸	✓	✓	✗ (Pretrained on human and mouse genomes only)	✗
DNABERT ¹⁹	✗ (Max context length of DNABERT-6 is 3072 bp, which is not sufficient to include a median length (26,288 bp) human protein coding gene ⁷⁰)	✓	✗ (Pretrained on human genome)	✓
Nucleotide Transformer ²⁰	✗ (Max context length is 6000 bp, which is not sufficient to include a median length (26,288 bp) human protein coding gene)	✓	✗ (Heavily biased towards human genome)	✓
HyenaDNA ²¹	✓	✓	✗ (Pretrained on Human genome)	✓
GenSLM-foundation model ²²	✗ (Single genes used for pretraining)	✓	✓	✓
GenSLM-SARS-CoV2 genome model ²²	✓	✓	✗ (fine-tuned on SARS-CoV2 genomes only)	✓

A method based on established ESM2

The work aims to build a genomic language model (gLM) based on an existing well-trained pLM (namely ESM2), in other words, to train the transformer-based gLM network to encode genomic information into gLM, using genomic contig data so that it may be used for downstream gene-related tasks. The method may be summarized as follows: A plural of sequences of 15~30 genes in contig is used as the initial training data. The gene sequences are converted into sequences of amino acids and then fed into the ESM2 encoder to yield protein embedding. The embedding is then masked and used as the input to the gLM. The gLM is trained to recover masked embedding. Initial results are provided about the potential use of pLM for integrating protein language models with genomic information.

We thank the reviewer for the kind feedback.

Experiments and Data Analysis

1. In Figure 3A and Figure 3B, UMAP is utilized to demonstrate the associative power of gLM. However, an excess of noise points may be obscuring the clarity of the visualization. Ideally, the UMAP representation should be consolidated rather than dispersed throughout the two-dimensional space. The authors may preprocess the data to remove outliers before visualizing with UMAP. More importantly, quantitative metrics, such as the KL divergence, should be provided to assess the differences or similarities between representations.

Thank you for the suggestion. We have removed these UMAP figures, and instead replaced them with quantitative measures of KL divergence between unannotated and annotated fractions of data, after removing outliers. The new analysis is added to L206:

“We compared the distributions of unannotated and annotated fractions of proteins in our dataset using context-free pLM embeddings and contextualized gLM embeddings. We found a statistically significant lower divergence between distributions of unannotated and annotated genes in gLM embeddings compared to pLM embeddings (paired t-test of Kullback-Leibler divergences, t-test statistic = 7.61, p-value < 1e-4, n = 10; see Methods for sampling and metric calculation).”

The method section corresponding to this calculation is pasted below:

“Distributions of unannotated and annotated embeddings

Distributions of unannotated and annotated embeddings in the database were compared using Kullback-Leibler (KL) divergence analysis. First, ten random samples of 10,000 subcontigs from the MGNify corpus. pLM and gLM embeddings of the genes were calculated using mean-pooled last hidden layer of ESM2 embeddings and mean-pooled last hidden layer of gLM respectively. Outliers were removed using Mahalanobis distance and a chi-squared threshold of 0.975. pLM and gLM embedding dimensions were reduced to 256 principal components (91.9 ± 1.72% and 80.1 ± 6.89% total variances explained respectively). KL divergence was calculated using the following equation:

$$D_{KL}(P||Q) = \frac{1}{2}(\text{tr}(\Sigma_1^{-1}\Sigma_0) - k + (\mu_1 - \mu_0)^T \Sigma_1^{-1}(\mu_1 - \mu_0) + \ln(\frac{\det \Sigma_1}{\det \Sigma_0}))$$

, where P corresponds to the distribution of unannotated genes and Q corresponds to the distribution of annotated genes, with μ_0, μ_1 respectively as means and Σ_0, Σ_1 respectively as covariance matrices. The significance of the KL divergence differences between pLM and gLM embeddings is calculated using a paired t-test across the ten samples.”

2. Comparative results with the base EMS2: While the paper provides an insightful discussion on contextualized gene embeddings, much of the evidence is presented qualitatively through visualizations. Specifically, experiments may be designed to quantitatively compare the performance of gLM embeddings against the original ESM2 embeddings. A quantitative comparison would provide a more rigorous and convincing assessment of the efficacy and relevance of the gLM embeddings in capturing meaningful genetic contexts.

Thank you for this feedback. We now accompany all UMAP visualizations with quantitative analysis. Below is the summary of all accompanying quantitative analyses.

Figure 2BFi: Silhouette scores to compare clustering according to labels. We also compare clustering with context-free pLM embeddings and contig-averaged pLM embeddings.

*Figure 4EF: We compare KL divergences and use embedding distance-based hierarchical clustering (**Extended Data 12**) to show clustering of newly identified TnsC-like AAA+ regulators with previously identified TnsC.*

Figure 5BC: We quantitate the prediction against chance simulation accuracy.

Figure 5DE: We compare the expressiveness of contig-averaged pLM and gLM embeddings for taxonomic classification by training a logistic regression classifier.

3. Results on enzyme function prediction: The authors have contrasted the results between pLM and pLM+gLM representations. A setup where only gLM is evaluated might provide insights into whether gLM holds comprehensive information for performing the downstream gene-related tasks effectively on its own, without the aid of pLM. It would further validate the inherent strengths and utility of the gLM embeddings.

*Thank you for the suggestion. We redid the EC number prediction analysis to incorporate the suggestions by all reviewers. The major changes are as follows: 1) We reduced the sequence identity threshold for train-test split to 30% to minimize leakage. 2) We compare predictor trained on pLM and gLM embeddings separately (instead of concatenated embeddings). Notably, in **Extended Data 8**, we compared pLM (context-free) embedding, gLM embedding where the*

queried protein is masked at the time of inference and therefore only context is utilized (context-only gLM embedding) and gLM embedding without masking (contextualized gLM embedding). In doing so, we show that masked gLM embedding, even without any pLM information, carries important information that can be leveraged for enzyme function prediction. 3) We conducted linear probing on different layers in gLM separately for EC number prediction and show that embeddings derived from layer 1 carries the most EC relevant information. 4) We added error estimates for linear probing results. Please find the updated methods section in L554-573:

And the updated results section and figures are below:

Contextualization improves enzyme function prediction

To test the hypothesis that genomic context of proteins can be used to aid function prediction, we compared how much the addition of context information can improve the expressiveness of protein representations for enzyme function prediction. First, we generated a custom MGY-EC dataset where the train and test data were split at 30% sequence identity for each EC class (see **Methods**). Second, we apply a linear probe (LP) to compare the expressiveness of representations at each gLM layer, with and without masking the queried protein (**Extended Data 8**). By masking the queried protein, we can assess gLM's ability to learn functional information of a given protein, only from its genomic context, without the propagation of information from the protein's pLM embeddings. We observed that a large fraction of contextual information pertaining to enzymatic function is learned in the first six layers of gLM. We also demonstrate that context information alone can be predictive of protein function, reaching up to $24.4 \pm 0.8\%$ accuracy. In contrast, without masking, gLM can incorporate information present in the context with the original pLM information for each queried protein. We observed an increase in expressivity of gLM embeddings also in the shallower layers, with accuracy reaching up to $51.6 \pm 0.5\%$ in the first hidden layer. This marks a $4.6 \pm 0.5\%$ increase from context-free pLM prediction accuracy (**Figure 3A**) and mean average precision (**Figure 3C**). Thus, we demonstrate that information that gLM learns from the context is orthogonal to information captured in pLM embedding. We also observed diminishing expressivity in enzyme function information with deeper layers of gLM; this reflects the masked pretraining objective that is independent of enzyme function prediction task and is consistent with previous examinations of LLMs, where specific layers perform better than others for downstream tasks. Finally, to further examine the expressiveness of these representations, we compared per-class F1 score gains (**Figure 3B**). We observe statistically significant differences in F1 scores (t-test, Benjamini/Hochberg corrected p-value < 0.05) between the two models in 36 out of 67 EC classes with more than ten samples in the test set. Majority (27 out of 36) of the statistical differences resulted in improved F1 score in MLP trained on gLM representations.

Figure 3. Contextualization of unannotated proteins. **A)** Linear probe EC classification accuracy for pLM(ESM2) representations and gLM (1st hidden layer) representations. **B)** F1-score comparisons of statistically significant (Benjamini/Hochberg corrected p -value < 0.05) differences in performance of pLM- and gLM-based EC number linear probes. EC classes are ordered with the largest gain with contextualization on the left to the largest loss with contextualization on the right. **C)** Precision-Recall curves of pLM- and gLM-based EC number linear probes. **D)** Histogram of variance (# bins = 100) calculated using contextualized embeddings (gLM; orange) and contig-averaged pLM (blue) embeddings of MGYPs that occur at least 100 times in the database. Histograms for unannotated and annotated fraction of the MGYPs are plotted separately and bars are not stacked. Annotated examples in the long right tail include phage proteins and transposases, reflecting their ability to self-mobilize (see annotations of top tens most variant genes in **Extended Data 9**).

A

B

D

Extended Data 8. Linear probing of context-free, context-only and contextualized gene embeddings. A) Schematics of how context-free, context-only, and contextualized gene embeddings are extracted for linear probing. Context-only gene embeddings are extracted by masking the queried gene, therefore the original pLM signal cannot propagate through to gLM representation. This can be used to quantify how gLM-learned context informs EC number

prediction independent of pLM signal. B) Schematics of how LP is used for EC number classification. C) Per-layer linear probing accuracy of gLM contextualized embeddings, where no-masking was performed at the time of inference. The 0th layer of gLM is equivalent to context-free pLM embedding. D) Per-layer linear probing accuracies of gLM context-only embeddings, where the queried genes are masked at the time of inference.

4. Regarding Figure 3(D): The current results show the outcomes based on contextualized embeddings. For a comprehensive understanding and fair comparison, results derived from context-free embeddings, in addition to those from the contextualized ones, would provide valuable information. This way can help readers assess the relative advantages or differences between the two methods more clearly.

Thank you for the suggestion. This comparison was also suggested by reviewer 3 and we have now updated Figure 3D to compare variances in contig averages of pLM embeddings with variances in gLM embeddings. Since pLM embedding is the same for each gene across all occurrences of the database, we took the averages of the pLM embeddings in the contigs for each occurrence of a gene and then compared the two distributions and the functions of the most context-variant proteins identified using contig-averaged pLM and gLM embeddings.

Soundness and Potential Impact

The paper claims to have demonstrated that the gLM model captures meaningful biological signals related to gene function, regulation, and taxonomy. However, providing results on how the gLM model has improved over the base ESM-2 quantitatively can be more direct and valuable. For example, in Figure 2, it would be informative to provide visualization of the context-free protein embeddings from ESM-2 to illustrate better what new insights are gained from gLM contextualization. In Figure 3C, comparing gLM to ESM-2 in EC number prediction would help benchmark performance gains.

Questions also arise about how much genomic knowledge has been embedded and whether the gLM can be called a gene model. The gLM model does not directly encode gene-related knowledge except using the gene-related contig data at the front end.

Therefore, the impact on the related field is limited due to insufficient experimental evidence.

Minor questions

1. While the number of genes in a contig can exceed 30, why was 30 chosen as the maximum value? Any experimental results to support the choice?

We chose 30 as the maximum context length because our corpus consists of metagenomic contigs, which are genomic fragments. Out of ~11M raw contigs with ≥ 15 genes in length, 67% of the contigs (consisting 40% of all genes in this dataset) were of ≤ 30 genes in length, therefore increasing the context length to > 60 genes would have resulted in many examples in our database to be $> 50\%$ padding, which would have reduced computational efficiency. We have added the contig length histogram as Extended Data X and elaborate on this design choice in the methods section in L476.

Extended Data 14. *Cumulative distribution of raw MGnify contig lengths (count of encoded genes), MGnify contigs < 15 genes were excluded from the corpus in this study. Red line is drawn at the contig length of 30 genes.*

2. In Figure 1b, the 'gLM' module is better described as the 'gLM encoder,' and the output may be a reconstructed representation.

Thank you for the suggestion. We have kept the diagram as it is to make it clear that gLM trained as illustrated in figure 1A is used for inference in figure 1B. We had considered designating gLM as gLM encoders in both 1A and 1B, but then we would also need to rename pLM as pLM encoder. The field uses the designation "pLMs" to effectively mean "pLM encoders", as they rarely have decoders trained, and therefore, for parallelism and simplicity, we have kept the original designation. However, we have updated the caption of Figure 1 to make clear that these refer to transformer encoders.

3. In Figure 5, parts (D) and (E): What UMAP parameters are chosen specifically for the two visualizations? It is better to provide the numbers for better interpretation.

Thank you for pointing this out, we used the same UMAP parameters ($n_neighbors = 15$, $min_dist = 0.1$) for all our visualizations and now state this in the methods section in L636-639.

Reproducibility

The code is provided. It should be easy to reproduce.

REVIEWER COMMENTS

Reviewer #2 (Remarks to the Author):

I appreciate the author's responses to my previous points. I have no more observations, least a typo in lines 884&885 for "unannotated genes" for both P & Q.

[Christian Dallago]

Reviewer #3 (Remarks to the Author):

I want to thank the authors for this thorough revision of the concerns raised by all the reviewers. I consider all my concerns being addressed.

Reviewer #5 (Remarks to the Author):

Thanks for the rebuttal from the authors. The author's response addressed most of my concerns, but a few questions remain:

1. The newly provided table (on both pages 3 and 19) that compares different gLMs on the Extended Data 1 is not that convincing in that the binary judgments ✓'s and ✗'s lack supporting evidence of numerical results. This question is concerned with the main contributions of the paper.
2. Regarding the chosen lengths to be between 15 to 30, it would be better to provide the performance results for different lengths in addition to the explanation of the reasons for the padding and computational efficiency.

Reviewer #2 (Remarks to the Author):

Responses

I appreciate the author's responses to my previous points. I have no more observations, least a typo in lines 884&885 for "unannotated genes" for both P & Q.

Thank you for spotting this, we have now fixed this typo.

[Christian Dallago]

Reviewer #5 (Remarks to the Author):

Thanks for the rebuttal from the authors. The author's response addressed most of my concerns, but a few questions remain:

1. The newly provided table (on both pages 3 and 19) that compares different gLMs on the Extended Data 1 is not that convincing in that the binary judgments ✓'s and ✗'s lack supporting evidence of numerical results. This question is concerned with the main contributions of the paper.

Thank you for this feedback. We have now added notes that provide supporting evidence for each cell.

Please find the new table below.

	Multi-gene input	Continuous representation of genes	Generalizable across organisms (Organism-agnostic pretraining)	Self-supervised language model
gLM (this study)	✓ (Max context of 30 genes)	✓ (Model inputs and outputs are continuous gene representations)	✓ (Metagenomic sequences with bias towards bacteria, archaea and viruses)	✓ (Masked language model)
pLMs ⁴⁻⁷ (e.g. ESM2, ProtBert, ProtT5, ProGen, CaLM)	✗ (Max context of 1 gene)	✓ (Continuous gene representation can be extracted from pooled intermediate layer)	✓ (ESM2 Trained on Uniref-50)	✓ (Masked language model)
Miller et al ¹⁶	✓ (Trained with max context of 5 genes)	✗ (Discrete gene vocab size of 563,589)	✓ (Trained on NCBI WGS and EBI MGnify, excluding green plants, fungi, and animals)	✗ (Based on word2vec, a one-layer word embedding model, not a language model)
Enformer ¹⁸	✓ (Max context 200 kbp)	✓ (Continuous gene representation can be extracted from pooled intermediate layer)	✗ (Pretrained on human and mouse genomes only)	✗ (Supervised with 5,313 human genomic tracks and 1,643 mouse genomic tracks)
DNABERT ¹⁹	✗ (Max context length of DNABERT-6 is 3072 bp, which is not sufficient to include a median length (26,288 bp) human protein coding gene ⁶³)	✓ (Continuous gene representation can be extracted from pooled intermediate layer)	✗ (Pretrained on human genome)	✓ (Masked language model)
Nucleotide Transformer ²⁰	✗ (Max context of 6000 bp, which is not sufficient to include a median length (26,288 bp) human protein coding gene)	✓ (Continuous gene representation can be extracted from pooled intermediate layer)	✗ (Heavily biased towards human genome)	✓ (Masked language model)

HyenaDNA ²¹	✓ (Max context of 1M bp)	✓ (Continuous gene representation can be extracted from pooled intermediate layer)	✗ (Pretrained on Human genome)	✓ (Causal language model)
GenSLM-foundation model ²²	✗ (Max context of 1 gene during pretraining)	✓ (Continuous gene representation can be extracted from pooled intermediate layer)	✓ (Pretrained on BV-BRC)	✓ (Causal language model)
GenSLM-SARS-CoV2 genome model ²²	✓ (Max context of 10,240 codons)	✓ (Continuous gene representation can be extracted from pooled intermediate layer)	✗ (Fine-tuned on SARS-CoV2 genomes only)	✓ (Causal language model)

2. Regarding the chosen lengths to be between 15 to 30, it would be better to provide the performance results for different lengths in addition to the explanation of the reasons for the padding and computational efficiency.

*Thank you for the suggestion. We have now added a supplemental figure **Extended Data 14A** (pasted below) where we illustrate increasing performance with increasing context length. Modeling performance is calculated using the accuracy of masked protein prediction on *E.coli* K-12's genome as described in the "Performance metric and validation" section of the methods and further elaborated in the legend. We also refer to this result in L478:*

*"We chose 30 as maximum context length because while longer context results in higher modeling performance (**Extended Data 14A**), 67% of the raw MGnify contigs with > 15 genes were of =< 30 genes in length (**Extended Data 14B**), and therefore increasing the context length beyond 30 would have resulted in many examples with padding (reduced computational efficiency)."*

Extended Data 14. A) Increasing model performance (estimated by absolute accuracy on *E. coli* K-12) with increasing context length (1 to 30 genes). See Methods section "Performance metric and validation" for detailed absolute accuracy calculation. Five random sets of genes in *E. coli* K-12 genome (# genes = 4315) were used for error estimation.

REVIEWERS' COMMENTS

Reviewer #5 (Remarks to the Author):

I have reviewed the authors' manuscript and responses. All my concerns have been properly addressed.

Reviewer #5 (Remarks on code availability):

None